# Evolution of neuronal anatomy and circuitry in two highly divergent nematode species

Ray L Hong[1,2†‡*], Metta Riebesell[1†], Daniel J Bumbarger[1§], Steven J Cook[3], Heather R Carstensen[2], Tahmineh Sarpolaki[1], Luisa Cochella[4], Jessica Castrejon[2], Eduardo Moreno[1], Bogdan Sieriebriennikov[1], Oliver Hobert[3,5], Ralf J Sommer[1]

[1]Department for Integrative Evolutionary Biology, Max-Planck Institute for Developmental Biology, Tuebingen, Germany; [2]Department of Biology, California State University, Northridge, Northridge, United States; [3]Department of Biological Sciences, Columbia University, New York, United States; [4]Research Institute of Molecular Pathology (IMP), Vienna Biocenter (VBC), Vienna, Austria; [5]Howard Hughes Medical Institute, Chevy Chase, United States

**Abstract** The nematodes *C. elegans* and *P. pacificus* populate diverse habitats and display distinct patterns of behavior. To understand how their nervous systems have diverged, we undertook a detailed examination of the neuroanatomy of the chemosensory system of *P. pacificus*. Using independent features such as cell body position, axon projections and lipophilic dye uptake, we have assigned homologies between the amphid neurons, their first-layer interneurons, and several internal receptor neurons of *P. pacificus* and *C. elegans.* We found that neuronal number and soma position are highly conserved. However, the morphological elaborations of several amphid cilia are different between them, most notably in the absence of 'winged' cilia morphology in *P. pacificus*. We established a synaptic wiring diagram of amphid sensory neurons and amphid interneurons in *P. pacificus* and found striking patterns of conservation and divergence in connectivity relative to *C. elegans*, but very little changes in relative neighborhood of neuronal processes. These findings demonstrate the existence of several constraints in patterning the nervous system and suggest that major substrates for evolutionary novelty lie in the alterations of dendritic structures and synaptic connectivity.

DOI: https://doi.org/10.7554/eLife.47155.001

**\*For correspondence:**
ray.hong@csun.edu

[†]These authors contributed equally to this work

**Present address:** [‡]Department of Biology, California State University, Northridge, Northridge, United States; [§]Allen Institute for Brain Science, Seattle, United States

## Introduction

Comparative studies on nervous system anatomy have a long tradition of offering fundamental insights into the evolution of nervous systems and, consequently, the evolution of behavior (*Schmidt-Rhaesa, 2007*). Traditionally, such comparative studies have relied on relatively coarse anatomical and morphometric comparisons. The relative simplicity of nematode nervous systems (*Schafer, 2016*) facilitates the determination and subsequent comparison of neuroanatomical features of distinct nematode species, thereby enabling an understanding of how members of the same phylum, sharing a common body plan, can engage in very distinct behaviors.

We examine here specific neuroanatomical features, from subcellular specializations to synaptic connectivity, of the nematode *Pristionchus pacificus* and compare these features with those of the nematode *Caenorhabditis elegans*. The species shared their last common ancestor around 100 million years ago, which is longer than the human-mouse separation (*Nei et al., 2001*; *Prabh et al., 2018*; *Rota-Stabelli et al., 2013*; *Werner et al., 2018*), and have since diverged to populate very discrete habitats and engage in distinct sets of behaviors. *C. elegans* is a free-living nematode that

**eLife digest** Nerve cells, also called neurons, are responsible both for sensing signals from the environment and for determining how organisms react. This means that the unique features of an animal's nervous system underpin its characteristic behaviors. Comparing the anatomy of the nervous systems in different animals could therefore yield valuable insights into how structural and behavioral differences emerge over time.

Behavioral variation often occurs even in similar-looking animals. One example is a group of microscopic worms, called nematodes. Although many nematode species exist, their overall body plans are the same, and the worms of each species contain a fixed number of cells. Despite these apparent similarities, different species of nematodes inhabit a variety of environments and may respond differently to the same signals.

The main sensory organs in nematodes are called the amphid sensilla. They are used to detect chemicals, as well as other inputs from the environment such as temperature and pheromones from other nematodes. Although researchers have often speculated that the number of cells in these organs and their arrangement are broadly the same across species, their anatomy had not been studied in detail.

Hong, Riebesell et al. compared the detailed structure and genetic features of the sensory systems in two distantly related species of nematode worms, *Pristionchus pacificus* and *Caenorhabditis elegans*. These two species behave in different ways, for example, *P. pacificus* is usually found in association with different species of beetles, while *C. elegans* is free-living and usually found on rotting fruit. By comparing the two, Hong, Riebesell et al. wanted to determine whether the diverse behaviors observed in the two species could be determined by differences between their sensory systems.

Experiments using electron microscopy yielded several thousand high resolution images spanning the entire sensory organ. These images were then used to create detailed reconstructions of the sensory nervous system in each worm species, demonstrating that both species had the same number of sensory nerve cells, allowing one-to-one comparisons between them.

Further analysis showed that while the overall structure of the neuronal connections remains the same between the two species, the neurons in *P. pacificus* made more diverse connections than those in *C. elegans*. Detailed studies of gene activity also revealed that neurons in each species switched on a slightly different group of genes, possibly indicating that each type of worm processes sensory signals in different ways.

These results shed new light on how nervous systems in related species can change over time without any change in neuron count. In the future, a better understanding of these changes could link the evolution of the nervous system to the emergence of different behaviors, in both simple and more complex organisms.

DOI: https://doi.org/10.7554/eLife.47155.002

can mainly be found in rotten fruit while members of the genus *Pristionchus* are regularly found in association with several species of scarab beetles, depending on geography (*Herrmann et al., 2006*; *Herrmann et al., 2007*; *Koneru et al., 2016*; *Ragsdale, 2015*). Additionally, *Pristionchus* nematodes are found in association with other soil arthropods, such as millipedes (*Yoshida et al., 2018*) and insect baits for soil-dwelling entomopathogenic nematodes (*Campos-Herrera et al., 2019*; *Kanzaki et al., 2018*), as well as other vegetal substrates (*Félix et al., 2018*). Species-specific entomophilic beetle association in *Pristionchus* is corroborated by several adaptations. First, *Pristion-chus* nematodes exhibit chemosensory responses towards insect pheromones and volatile plant compounds (*Hong and Sommer, 2006*; *Hong et al., 2008*; *Cinkornpumin et al., 2014*). Second, *P. pacificus* dauer larvae secrete a high molecular weight wax ester that promotes collective host finding (*Penkov et al., 2014*). Finally, *Pristionchus* species show predatory behavior towards *C. elegans* and other nematodes (*Bento et al., 2010*; *Liu et al., 2018*). All these behavioral features likely require substantial modifications in the nervous system.

One obvious potential substrate for evolutionary adaptions to distinct ecological habitats and interactions with other species is the perception and processing of sensory information. The amphid

sensilla, comprised of a pair of bilaterally symmetrical sensilla in the head that are open to the external environment, are the largest nematode chemosensory organs (*Bargmann, 2006*; *Bargmann and Horvitz, 1991*; *Bargmann et al., 1993*). In the model organism *C. elegans*, the anterior sensilla include the amphid sensilla, twelve inner and six outer labial neurons, and four cephalic neurons, along with their associated sheath and socket support cells (*Ward et al., 1975*; *Ware et al., 1975*). There are also additional sensory receptors without glia broadly grouped into non-ciliated (URX, URY, URA, URB) and ciliated (BAG, FLP) neurons that are involved in gas sensing, mechanoreception, and male mate-searching behavior (*Barrios et al., 2012*; *Chatzigeorgiou and Schafer, 2011*; *Doroquez et al., 2014*; *Gray et al., 2004*; *Hallem and Sternberg, 2008*; *Hallem et al., 2011*; *Perkins et al., 1986*). Comparisons of the *C. elegans* amphid neuroanatomy to those of free-living and parasitic nematodes such as *Acrobeles complexus* (*Bumbarger et al., 2009*), *Haemonchus contortus* (*Li et al., 2000*; *Li et al., 2001*), and *Parastrongyloides trichosuri* (*Zhu et al., 2011*) have shown that the number and arrangement of the amphid neurons are broadly conserved. However, a more fine-grained comparative analysis of distinct sensory structures as well as their connection to downstream circuits has been largely lacking so far.

Using the 3D reconstructions of serial thin section transmission electron microscopy (TEM), we describe here detailed features of the sensory anatomy of *P. pacificus* as well as the synaptic wiring of sensory neurons to their main, postsynaptic interneurons. Comparing anatomical features of sensory circuitry, from ciliated sensory endings to soma and axon position to synaptic connectivity, we reveal striking patterns of similarities and dissimilarities. The most striking similarity is the overall conservation of neuronal soma and process positioning while the most striking patterns of divergences lie in fine structural details of sensory anatomy as well as synaptic connectivity.

## Results

### Overall amphid architecture

Using a combination of 3D reconstructions from TEM sections of two young adult hermaphrodites, as well as live dye uptake and transgene reporter analysis, we set out to characterize the amphid sensory circuitry of *P. pacificus* in order to undertake a comparative analysis with the amphid sensory circuitry of *C. elegans*. For comparison with *C. elegans*, we used electron micrographs and findings from both legacy (*White et al., 1986*) and modern EM methodologies (*Doroquez et al., 2014*). Despite being approximately 40 years old, the EMs used by John White and colleagues to create The Mind of a Worm remain the most complete publicly available data of the adult hermaphrodite *C. elegans* nervous system. While the methods used to create these legacy EM series are technologically inferior to current practices (chemical fixation, analog microscopy, thicker sections), the overall staining and elucidation of synaptic zones remain useful and valuable for anatomical comparisons as has been shown in a recent publication on whole-animal connectomes of both *C. elegans* sexes (*Cook et al., 2019*). Recent studies of the amphid dendritic endings in *C. elegans* using the modern High Pressure Freezing (HPF) method have validated the original studies, resulting in a richer description of ultrastructural anatomy (*Doroquez et al., 2014*). The numerous anatomical similarities and differences we observed across species are both reproducible and share equivalences to previous nematode comparative anatomical studies.

To account for all the amphid neurons in *P. pacificus*, we identified and traced every amphid neuron from the tip of its cilium in the channel near the mouth to its cell body posterior to the nerve ring in two young adult hermaphrodites. Like *C. elegans*, *P. pacificus* possesses 12 neurons per amphid sensillum. All amphid neurons and the amphid sheath cell (AMsh) in *P. pacificus* have their cell bodies in the lateral ganglion posterior to the nerve ring, which resembles the condition found in *C. elegans* and other nematodes. At the anterior end, the tips of the dendrites are housed by the processes of a pair of amphid sheath cells, which are glial cells that connect to the amphidial pore in the cuticle via a pair of amphid socket cells (AMso) that expose the amphid neuronal cilia to the environment (*Figure 1A–D*). All amphid neurons have ciliated endings with a circle of varying numbers of doublet microtubules surrounding a few inner singlet microtubules in their transition zones and middle segments (*Figure 1F*). We counted 13 cilia (from 11 neurons) in each amphid channel in *P. pacificus,* compared to 10 cilia (from eight neurons) in the channel in *C. elegans* (*Figure 1D–F*, *Table 1*). The greater number of ciliated endings in *P. pacificus* coincides with a conspicuous lack of

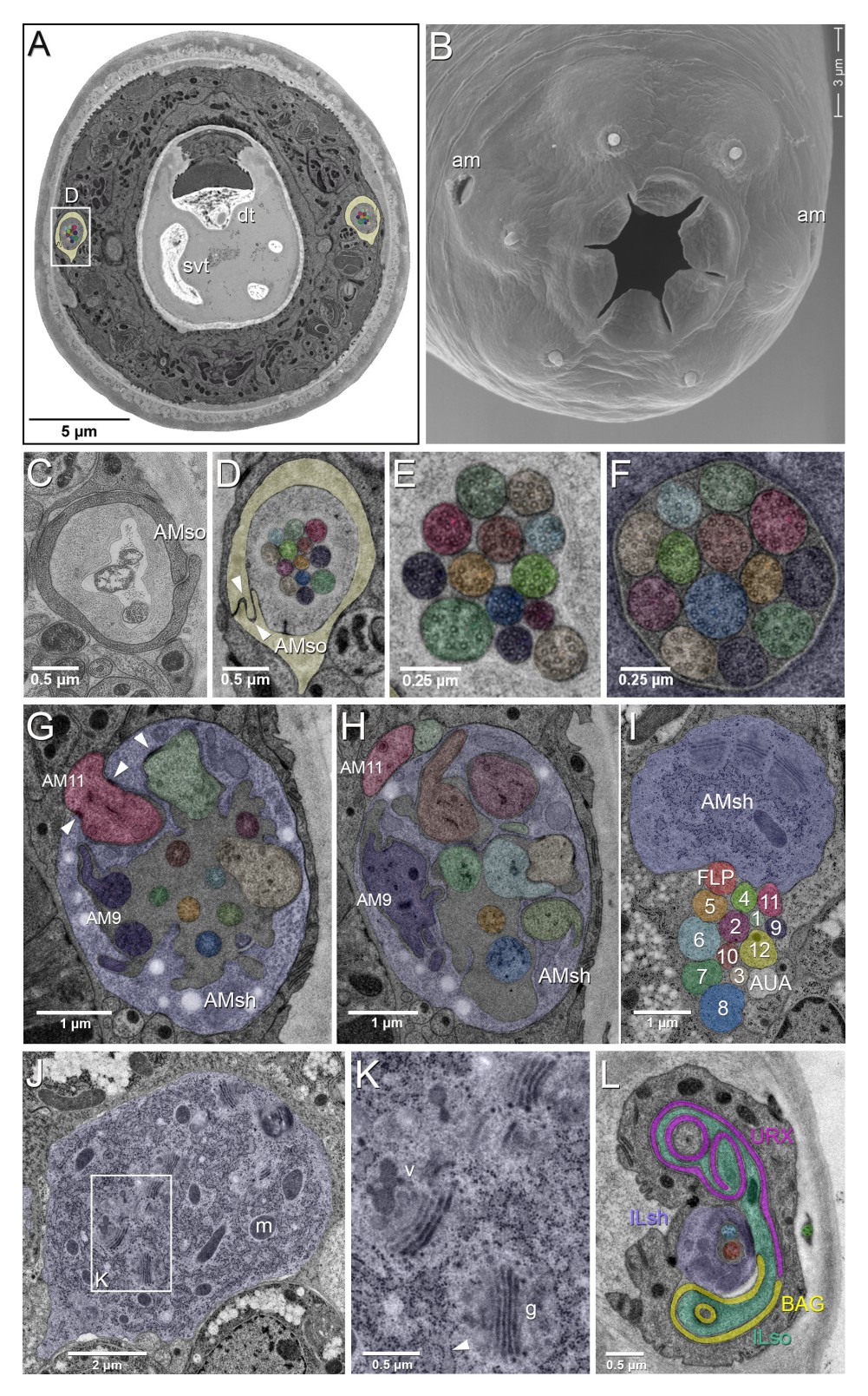

**Figure 1.** EM of the amphid sensillum and two other sensory neurons of *P. pacificus* hermaphrodite adults. All images are from specimen 107 (Series 14), except B and C. As specimens were sectioned from the head, left structures appear on the right side in the images and vice versa. (**A**) Complete transverse section 6.8 μm from the tip of the head showing the amphid socket cells (light yellow) with false-colored amphid cilia bundles inside the channels. Note the dorsal tooth (dt) and sub-ventral tooth (svt) in the buccal cavity. (**B**) Scanning electron micrograph of the head of an adult animal

*Figure 1 continued on next page*

*Figure 1 continued*

showing the two amphid openings (am). (**C–H**) Transverse TEM sections through the amphid cilia at various levels from anterior to posterior (**C**) Left amphid channel close to the pore, containing the tips of the longest three cilia; the channel is formed by the amphid socket cell (AMso). (Specimen DB-9–1) (**D**) Right amphid channel slightly more posterior than in A (7.1 µm from the tip of the head) with all 13 cilia visible in the channel matrix. Arrowheads indicate the autocellular junction of the amphid socket cell (AMso). (**E**) Distal segments of the amphid cilia with singlet microtubules in the left channel 6.85 µm from the tip. (**F**) Middle segments of the right amphid cilia with discernable nine outer doublet microtubules that make up the core axoneme 11.25 µm from the tip; here the channel is formed by the amphid sheath cell (lilac). (**G**) Section through the region of sheath entry 13.1 µm from the tip, showing the thick periciliary membrane compartment (PCMC) of AM11 (red) entering the left amphid sheath cell (AMsh), the green and beige PCMCs have just completed their entry. Additionally, the sheath lumen harbors the basal parts of the double cilia of AM9 (dark lilac) with typical transition zone (TZ) arrangement of microtubules and seven further TZ cilia. Arrowheads mark the adherens junctions between the PCMCs and the amphid sheath. (**H**) A slightly more posterior section through the left sheath cell (13.85 µm from the tip), showing the distinctively large and irregular outline of the PCMC of AM9 (dark lilac) and the PCMCs of 6 other dendrites. The base of the AM8 cilium (blue) and the TZ of AM5 (orange) are found in the lumen. AM11 is still outside the sheath. (**I–K**) Transverse TEM sections through the posterior AMsh process and cell body (lilac). (**I**) Right-side amphid nerve directly anterior of the nerve ring (88.3 µm from the tip) consisting of the AMsh process (sh) with prominent Golgi stacks and the 12 amphid neuron dendrites. AM1 to AM12 are labeled by number and color. The color code is used throughout the paper. FLP (reddish, just below sheath) and AUA (white) join the amphid process bundle until they diverge from it or end. (**J**) AMsh cell body (100.8 µm from tip) with numerous mitochondria (**m**) and Golgi stacks. (**K**) A higher magnification of the Golgi apparatus (g) and vesicles (v) in the midst of ribosome-studded rough ER cisternae (arrowhead). (**L**) The bilaterally symmetrical URX and BAG neurons have extended flattened ciliary endings associated with the lateral Inner Labial socket cell (ILso) process; 2.05 µm from the tip. Comparable *C. elegans* EM annotations are available at SlidableWorm http://www.wormatlas.org/SW/SW.php/. See *Figure 1—figure supplements 1–3*, and *Figure 1—videos 1–3* for details on the URX, URY, URA, and BAG neurons.

DOI: https://doi.org/10.7554/eLife.47155.003

The following video and figure supplements are available for figure 1:

**Figure supplement 1.** URX is ciliated.
DOI: https://doi.org/10.7554/eLife.47155.004
**Figure supplement 2.** URY, URX, URA and BAG endings.
DOI: https://doi.org/10.7554/eLife.47155.005
**Figure supplement 3.** A comparison of select non-amphid ciliated neurons in three nematode species.
DOI: https://doi.org/10.7554/eLife.47155.006
**Figure 1—video 1.** Video of URX 3D reconstruction.
DOI: https://doi.org/10.7554/eLife.47155.007
**Figure 1—video 2.** Video of URY 3D reconstruction.
DOI: https://doi.org/10.7554/eLife.47155.008
**Figure 1—video 3.** Video of URA 3D reconstruction.
DOI: https://doi.org/10.7554/eLife.47155.009

neurons with winged dendritic morphology found in the *C. elegans* wing cells (AWx), whose elaborate ciliary endings terminate as invaginations inside the distal amphid sheath cell cytoplasm rather than in the amphid channel. This indicates that the AWA, AWB, AWC cellular homologs in *P. pacificus* are among the 11 single or double ciliated neurons in the channel and, therefore, do not display a characteristic 'wing'-shaped cilia morphology. The only other known amphid neurons with elaborate dendritic processes resembling the AWA cells are found in the swine parasite, *Oesophagostomum dentatum* (*Hoholm et al., 2005*). Because the wing neurons are the most morphologically distinct and best studied amphid neurons in *C. elegans*, their absence in most other nematodes contributes to the difficulty in assigning homology in the amphid neurons in these other nematode species (*Ashton et al., 1995*; *Bumbarger et al., 2007a*; *Bumbarger et al., 2009*; *Li et al., 2001*; *Ward et al., 1975*).

We designated provisional names for the *P. pacificus* amphid neurons utilizing 'AM' for amphid, followed by numbers 1 to 12 (*Table 2*). Of the 11 dendrites in the channel, nine have single ciliated endings and only two of them, AM3 and AM9, possess dual-ciliated endings. The lack of neurons with winged morphology compelled us to consider several other criteria for assigning amphid neuron homology between *P. pacificus* and *C. elegans*, such as i) relative cell body positions, ii) axon projections, iii) manner of dendrite entry into the amphid sheath cell, iv) number of cilia in channel, v) DiI dye filling properties, and vi) connections to first layer interneurons. While no single criterion allows homology assignment for all amphid neurons, our analysis results in high confidence homology assignments for most neurons. Using these provisional homologies, we proceeded to evaluate possible evolving features and found conservation and divergence in synaptic connectivity and

**Table 1.** Comparison of amphid neurons in various nematode species.

The total number of neurons with dendritic processes encased in the sheath cell in a single amphid compartment is indicated. These neurons are further categorized as neurons with dendrites having single or double ciliated endings in the amphid channel, or as specialized 'wing' neurons with endings outside the channel but within the sheath cell other than the finger cell. The ASC neurons in L1 larvae of *P. trichosuri* described by *Zhu et al. (2011)* match this criterion but lack the ciliary elaborations known from *C. elegans* wing neurons.

| Species | Total neurons | Dendritic ends in channel | Cilia in channel | | | Wing |
| | | | Total count | Single | Double | Neurons |
| --- | --- | --- | --- | --- | --- | --- |
| *Pristionchus pacificus* | 12 | 11 | 13 | 9 | 2 | 0 |
| *Caenorhabditis elegans*[a] | 12 | 8 | 10 | 6 | 2 | 3 |
| *Haemonchus contortus*[b] | 12 | 10 | 13 | 7 | 3 | 0 |
| *Strongyloides stercoralis*[c] | 13 | 12 | 12 | 12 | 0 | 0 |
| *Parastrongyloides trichosuri*[d] | 13 | 11 | 11 | 11 | 0 | 1? |
| *Acrobeles complexus*[e] | 13 | 12 | 12 | 10 | 1 | 0 |

a. *Ward et al. (1975)*
b. *Li et al. (2001)*
c. *Ashton et al. (1995)*
d. *Zhu et al. (2011)*
e. *Bumbarger et al. (2007a)* and *Bumbarger et al. (2009)*
DOI: https://doi.org/10.7554/eLife.47155.010

orthologous transcriptional reporters. At the end of this analysis, we will revisit homology assignments and will discuss potential strategies to overcome current limitations.

## The finger neuron

The most unambiguous amphid neuron homolog found in *P. pacificus* is AM12, with finger-like dendritic endings akin to the finger-neuron in most other nematode species: *C. elegans, Ancylostoma caninum, Oesophagostomum dentatum, Haemonchus contortus, Acrobeles complexus* and *Parastrongyloides trichosuri* L1 (*Bhopale et al., 2001*; *Bumbarger et al., 2009*; *Hoholm et al., 2005*; *Li et al., 2001*; *Ward et al., 1975*; *Zhu et al., 2011*) (*Figures 2A, C* and *3A*, *Figure 2—figure supplements 1* and *2*). The sensory ending of the AM12 dendrite is formed by a short cilium of about 1 μm length (in *C. elegans* about 500 nm, see Figure 12A in *Doroquez et al., 2014*) and a complex of 30–40 microvilli-like projections branching off from the periciliary membrane compartment (PCMC), which strongly resembles the morphology of the AFD neurons responsible for thermosensation in *C. elegans, A. caninum,* and *H. contortus* (*Bhopale et al., 2001*; *Li et al., 2001*; *Mori and Ohshima, 1995*). Similar to AFD endings in other species, the AM12 cilium is fully embedded in the amphid sheath cell process, does not enter the lumen of the sheath cell or the amphid channel and thus has no contact to the outside environment. We also observed that like in other species, the cell body of AM12(AFD) is the most anterior among the amphid neurons, located just ventral of the lateral midline (*Figure 3C*). Nevertheless, there is a small difference compared to other species – the AM12 sensory ending is located in a part of the AMsh that is clearly posterior to the lumen into which the other amphid dendrites enter. Taken altogether, we conclude that AM12 is the *P. pacificus* homolog of AFD. The confirmation of the AM12(AFD) neurons' role in thermosensation, however, will ultimately depend on cell ablation experiments followed by behavioral assays.

Thus, of the six nematode species with detailed descriptions of their amphidial sensory neuroanatomy (*P. pacificus, C. elegans, H. contortus, S. stercoralis, A. complexus, Parastrongyloides trichosuri* L1) the only nematode species known so far not to share finger-like dendritic endings is the mammalian parasite *Strongyloides stercoralis*, which has instead evolved a lamellar morphology for its putative thermosensory neuron ALD (*Ashton et al., 1995*; *Bumbarger et al., 2007a*; *Bumbarger et al., 2009*; *Li et al., 2001*; *Ward et al., 1975*; *Zhu et al., 2011*).

**Table 2.** Provisional nominations of putative amphid neuronal homologs between *C. elegans* and *P. pacificus*. Morphological data based on TEM series of specimens 107 and 148. Features supporting cellular homology include axon projections, lipophilic dye uptake, cell body positions, and chemical synapses to the four first layer amphid interneurons (AIA, AIB, AIY, AIZ) or the AUA. DiI filling neurons in *C. elegans* are ASI, ASJ, ASK, ADL, ASH, AWB (weak), *ADF (shows only weak FITC uptake, not DiI). DiO filling neurons in *C. elegans* are ASI, ASJ, ASK, ADL, ASH, AWB. PCMC: periciliary membrane compartment. Structures of *C. elegans* amphid neurons are available at http://www.wormatlas.org/images/NeuronImageList.jpg.

| *P. pacificus* neuron | Axon termination site in nerve ring | Dye filling | Feature(s) supporting homology | Likely *C. elegans* homolog | Main difference compared to *C. elegans* |
|---|---|---|---|---|---|
| AM1 | dorsal midline | DiI only | DiI uptake, cell position, cell body morphology | ASH | lack DiO uptake |
| AM2 | dorsal midline | DiI + DiO | DiI + DiO uptake; branched axon not through commissure; AIA/AIB | ADL | single ciliated vs. dual ciliated |
| AM3 | dorsal midline | none | none | AWA | dual ciliated vs. wing; *Ppa-odr-3* expression |
| AM4 | dorsal midline | DiI only | DiI uptake; AIA/AIB | ASK | lack of DiO uptake; *Ppa-odr-3* expression; |
| AM5 | dorsal midline | none | *Ppa-che-1* expression | ASE | axons cross the midline in *C. el.* |
| AM6 | lateral midline | none | short axons;cell position | ASG | *Ppa-che-1* expression |
| AM7 | cross dorsal midline | none | axons overlap dorsally; dorsal sheath entry; cell position; AIA/AIB/AIY/AIZ | AWC | single ciliated vs. wing; *Ppa-odr-7* expression |
| AM8 | dorsal midline | DiI + DiO | DiI and DiO uptake; ventral sheath entry;cell position | ASJ | none |
| AM9 | dorsal midline | DiI only | dual ciliated; DiI uptake*; prominent PCMC; connect to AUA | ADF | *Ppa-odr-7* expression |
| AM10 | dorsal midline | none | AIA/AIB/AIY/AIZ | ASI | lack DiI and DiO uptake |
| AM11 | dorsal midline | DiI only | weak DiI uptake; cell position; AIA/AIB/AIY/AIZ | AWB | single ciliated vs. wing; lack DiO uptake |
| AM12 | dorsal midline | none | finger dendrite morphology; cell position; first to enter the AMsh | AFD | more posterior position in AMsh due to lack of winged neurons |
| AMU1 | dorsal midline | none | short 'dendritic' process; cell position; exclusive output of ADF | AUA | none |

DOI: https://doi.org/10.7554/eLife.47155.011

## Cell body positioning and axonal projections of other amphid neurons

Without other signature dendritic endings to nominate possible amphid homologs, we turned to likely conservations in cell body position as well as in the projection trajectories of individual axon processes that enter and terminate in the nerve ring. Using the 3D reconstructions of amphid neurons from EM sections (specimen 107, 148; *Figure 3A*; *Figure 3—videos 1–15*), we first looked for *P. pacificus* neurons that might share another defining feature of the *C. elegans* AWC neurons: the AWC axons cross the dorsal midline, overlap each other and terminate just before reaching the lateral midline. We identified only one pair of single ciliated amphid neurons that shares this feature, the AM7 (*Table 2*). The AM7(AWC) cell body is located between the AM1 and AM8 neurons, which are likely the respective cellular homologs of ASH and ASJ based on conservation in DiI uptake and cell body positions along the ventral edge of the amphid neuron cluster (discussed below). As the AM7 and AWC axons both cross the dorsal midline and terminate above the lateral midline, we regard them as homologs based on this singular feature, although the AM7(AWC) cell body appears to have shifted from a position ventral of ASH in *C. elegans* to a position between AM1(ASH) and AM8(ASJ) in *P. pacificus*, such that the three cell types are just ventral and parallel to the lateral midline on each side.

Two other amphid neurons also show strong resemblance to their *C. elegans* counterparts according to their unique axon projections. The *P. pacificus* AM6 is likely the homolog of the *C. elegans* ASG neuron because these counterparts have the unique property of short axon projections that do not project into the nerve ring much further than the lateral midline, terminating before the

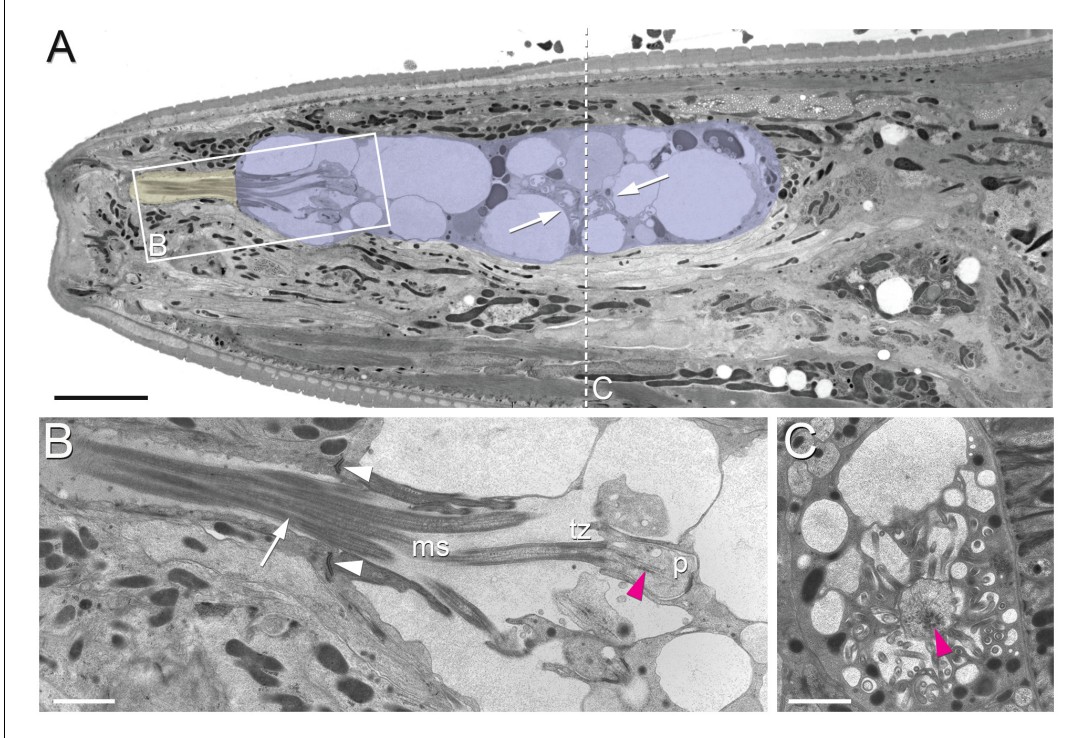

**Figure 2.** Position of amphid cilia in AMso and AMsh. (A–B) Sagittal TEM sections of a *P. pacificus* young adult hermaphrodite. (A) The yellowish shading highlights the narrow amphid channel, which is formed by the socket cell process and contains the distal segments of the 13 amphid cilia. The lilac shading highlights the expanded region of the amphid sheath cell process, which harbors the proximal segments of the amphid cilia in its anterior lumen and, more posteriorly, the finger cell in its cytoplasm amidst a multitude of vesicles. Arrows point at the space taken up by the finger cell cilium with its projections. Scale bar: 5 μm. (B) Detail of A (from a neighboring section) showing the different ciliary regions: middle segment (ms), transition zone (tz), and periciliary membrane compartment (p). The arrow points at cilia in the narrow, matrix-filled amphid channel. Arrowheads mark the adherens junctions between the AMsh and AMso processes. The red arrowhead indicates the ciliary rootlet within the periciliary membrane compartment (p) of one of the amphid neurons. Scale bar: 1 μm. (C) Transverse section through the periciliary membrane compartment of the amphid finger cell with central rootlet (red arrow head) and numerous fingerlike villi in various orientations. The dotted line in A represents the approximate plane of sectioning. Scale bar: 1 μm. The order of each dendrite entry is shown in *Figure 2—figure supplements 1* and *2*. Comparative sagittally sectioned *C. elegans* nose images are present at http://www.wormatlas.org/hermaphrodite/neuronalsupport/jump.html?newLink = mainframe.htm and newAnchor = Amphidsensilla31 (Figure 35). *Figure 2—figure supplements 1* and *2*.

DOI: https://doi.org/10.7554/eLife.47155.012

The following figure supplements are available for figure 2:

**Figure supplement 1.** Posterior-anterior path of amphid dendrite entry into the left sheath glia of specimen 107.

DOI: https://doi.org/10.7554/eLife.47155.013

**Figure supplement 2.** Posterior-anterior path of amphid dendrite entry into the right sheath glia of specimen 107.

DOI: https://doi.org/10.7554/eLife.47155.014

dorsal midline. The *P. pacificus* AM2 is the likely homolog of the *C. elegans* ADL neuron because like in ADL, the AM2 axons are the only amphid sensory neurons that do not run through the amphidial commissure but enter the nerve ring directly from an anterior projection before branching in the dorsal-ventral direction, which is unique among all amphid neurons in both species (*Table 2*). If axon projection is more highly conserved than cilia branching, then the *P. pacificus* cellular homolog for ADL is single ciliated and not double ciliated. The nomination of AM2 as the ADL homolog is corroborated by its cell body position just posterior to AM4 and its ability to take up DiI. Altogether, the conserved unique axon trajectories of AM2, AM6, and AM7 argue they are the likely homologs of the *C. elegans* ADL, ASG, and AWC neurons, respectively.

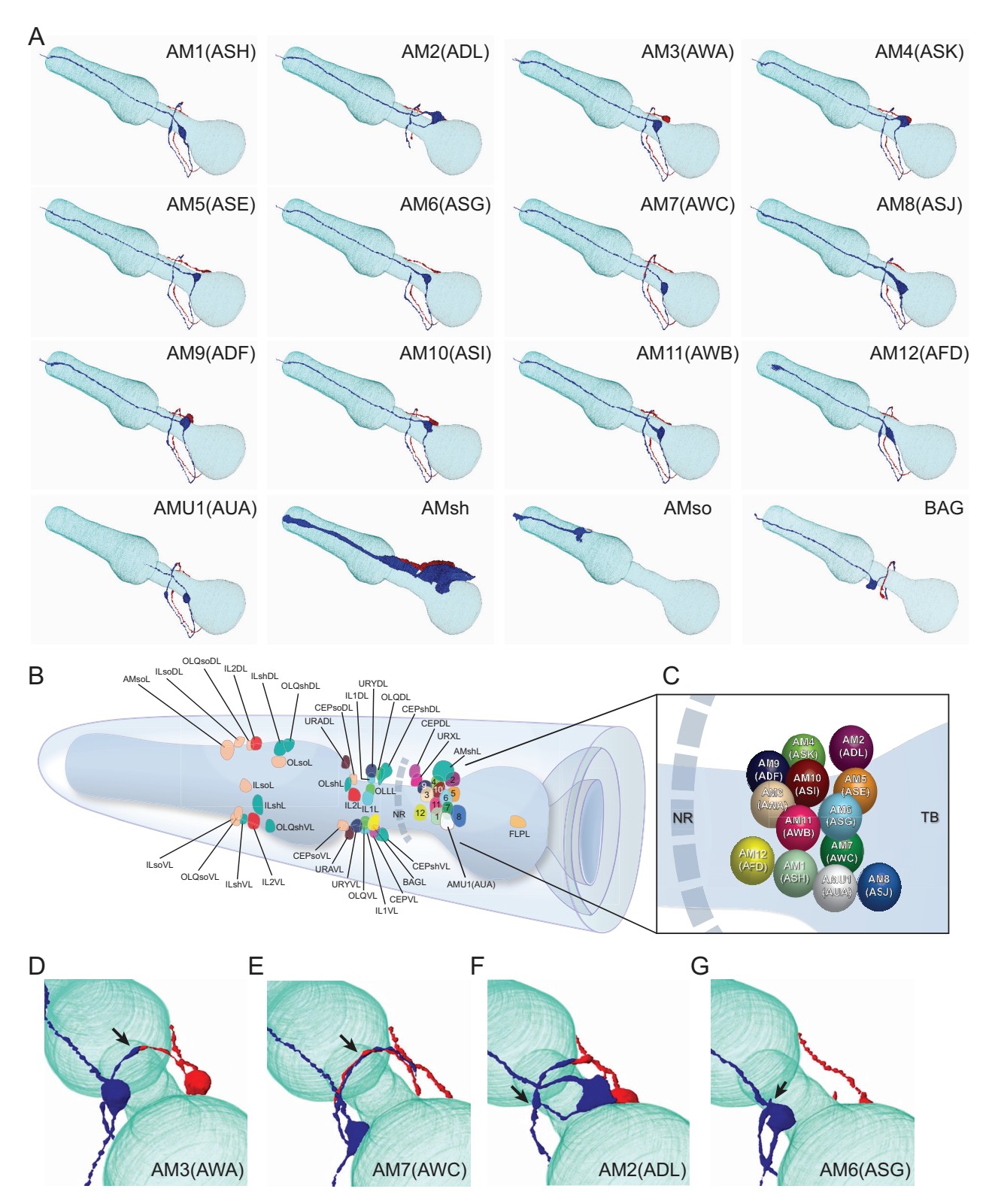

**Figure 3.** Overview of the amphid sensilla. (**A**) Three-dimensional renderings of individual pairs of amphid neurons, amphid sheath, amphid socket, internal sensory receptor AUA, and BAG neuron in a *P. pacificus* young adult hermaphrodite (Specimen 107). Left lateral view, anterior is to the left. Blue and red renderings denote left and right counterparts, respectively. The pharynx is outlined in teal. Nomenclature is displayed with *P. pacificus* first and *C. elegans* name in parenthesis. Black arrows denote lateral nerve ring entry (not through amphid commissure) and branched axons in AM2

*Figure 3 continued on next page*

*Figure 3 continued*

(ASG), short axons ending at the lateral midline in AM6(ASG), dorsal overlap of axon endings in AM7(AWC), fingerlike endings of AM12(AFD) dendrite terminating within the sheath cell posterior to the nose, and short dendritic-like endings of AUA anterior to the nerve ring *Figure 3—videos 1–15*. (B) Schematic diagram of the positions of the nuclei of the left cuticular sensilla with sheath and socket cells and of the internal receptors. The nerve ring (NR) is indicated by a dotted line. Amphid cell bodies are labeled 1–12 by their last number, with color code the same as in *Figure 1*. (C) Schematic of amphid nuclei as seen from the left side with full names, nuclei slightly enlarged. (TB = terminal bulb). (D–G) The four different types of axon projections in *P. pacificus* amphid neurons: right and left axons: (D) meet and end with very little overlap at the dorsal midline (all except AM6/ASG and AM7/AWC, also AMU1/AUA) (AM3 is shown to represent this group); (E) cross extensively over the dorsal midline (AM7/AWC); (F) branch laterally into a dorsal and a ventral process due to lateral entry into the NR instead of the usual ventral entry of all other amphid neurons that come from the amphid commissure (AM2/ADL); (G) short axons end at the lateral midline (AM6/ASG). Cartoons of *C. elegans* amphid neuron anatomy are present at http:// www.wormatlas.org/images/NeuronImageList.jpg. Cartoons of *C. elegans* ganglia are present at http://www.wormatlas.org/images/VCMNganglia.jpg. 3D renderings of individual neurons are shown in *Figure 3—videos 1–15*.

DOI: https://doi.org/10.7554/eLife.47155.015

The following videos are available for figure 3:

**Figure 3—video 1.** 3D rendering of AM1(ASH) neurons.
DOI: https://doi.org/10.7554/eLife.47155.016
**Figure 3—video 2.** 3D rendering of AM2(ADL) neurons.
DOI: https://doi.org/10.7554/eLife.47155.017
**Figure 3—video 3.** 3D rendering of AM3(AWA) neurons.
DOI: https://doi.org/10.7554/eLife.47155.018
**Figure 3—video 4.** 3D rendering of AM4(ASK) neurons.
DOI: https://doi.org/10.7554/eLife.47155.019
**Figure 3—video 5.** 3D rendering of AM5(ASE) neurons.
DOI: https://doi.org/10.7554/eLife.47155.020
**Figure 3—video 6.** 3D rendering of AM6(ASG) neurons.
DOI: https://doi.org/10.7554/eLife.47155.021
**Figure 3—video 7.** 3D rendering of AM7(AWC) neurons.
DOI: https://doi.org/10.7554/eLife.47155.022
**Figure 3—video 8.** 3D rendering of AM8(ASJ) neurons.
DOI: https://doi.org/10.7554/eLife.47155.023
**Figure 3—video 9.** 3D rendering of AM9(ADF) neurons.
DOI: https://doi.org/10.7554/eLife.47155.024
**Figure 3—video 10.** 3D rendering of AM10(ASI) neurons.
DOI: https://doi.org/10.7554/eLife.47155.025
**Figure 3—video 11.** 3D rendering of AM11(AWB) neurons.
DOI: https://doi.org/10.7554/eLife.47155.026
**Figure 3—video 12.** 3D rendering of AM12(AFD) neurons.
DOI: https://doi.org/10.7554/eLife.47155.027
**Figure 3—video 13.** 3D rendering of AMU1(AUA) neurons.
DOI: https://doi.org/10.7554/eLife.47155.028
**Figure 3—video 14.** 3D rendering of Amphid sheath (AMsh).
DOI: https://doi.org/10.7554/eLife.47155.029
**Figure 3—video 15.** 3D rendering of Amphid socket (AMso).
DOI: https://doi.org/10.7554/eLife.47155.030

## The presumptive taste receptor neuron pair ASE in *P. pacificus* may not be functionally lateralized

In *C. elegans,* the axons of the ASEL and ASER neuron pair cross the dorsal midline around the nerve ring until they end ventrally close to the entry point of the contralateral axon (*White et al., 1986*). In *P. pacificus*, the cell bodies of AM5 are located just ventral to those of the AM2(ADL) neurons at the second-most posterior position of the amphid cluster in the lateral ganglion [AM8(ASJ) is the most posterior pair]. The conservation of cell body position and the expression of the *Ppa-che-1p::rfp* transgene reporter (see below) provide independent support to nominate the AM5 neurons as the ASE homologs. However, the *P. pacificus* AM5(ASE) axons do not cross each other at the dorsal midline, but rather terminate at the dorsal midline. At their respective termination points *P. pacif- icus*, ASEL and ASER make a small gap junction with each other, like all but two amphid sensory

neuron pairs do (*Table 2*). This apparent electrical coupling between the *P. pacificus* ASE homologs is notable because in *C. elegans* no such gap junctions are formed between the ASEs (*White et al., 1986*). The consequent lack of electrical coupling between the *C. elegans* ASEL and ASER neurons has been found to be necessary to produce a physiological asymmetry between these neurons, such that both neurons are differentially activated by distinct sensory cues, that is their function is 'lateralized' ('left/right asymmetric') (*Pierce-Shimomura et al., 1999*; *Ortiz et al., 2009*; *Suzuki et al., 2008*). Moreover, it has been shown that artificial establishment of gap junctions between the left and right *C. elegans* ASEs leads to loss of functional lateralization and changes in chemotaxis behavior (*Rabinowitch et al., 2014*). The apparent coupling of ASEL and ASER in *P. pacificus* suggests that these two neurons are not functionally lateralized in *P. pacificus*.

Two additional genetic observations are consistent with a lack of functional lateralization in *P. pacificus*: First, the key regulatory factor that triggers the asymmetry of the ASEL/R neurons in *C. elegans*, the miRNA *lsy-6* (*Cochella and Hobert, 2012*; *Johnston and Hobert, 2003*), does not exist in the *P. pacificus* genome and is apparently specific to the *Caenorhabditis* crown clade (*Ahmed et al., 2013*). Second, the ASEL and ASER neurons in *C. elegans* (as well as closely related *Caenorhabditis* species) each express a different subfamily of duplicated and chromosomally-linked, receptor-type guanylyl cyclases (rGCYs), the ASER-rGCYs (e.g. *gcy-1, gcy-2, gcy-3, gcy-4, gcy-5*) and the ASEL-rGCYs (*gcy-6, gcy-7, gcy-14, gcy-20*) (*Ortiz et al., 2006*; *Yu et al., 1997*), several members of which are thought to be salt chemoreceptors (*Ortiz et al., 2009*). In contrast, the *P. pacificus* genome contains no *C. elegans*- ASEL-type or ASER-type rGCYs (*Figure 4—figure supplements 1* and *2*). Together with the changes in electrical coupling of the ASE neurons in *C. elegans* versus *P. pacificus*, the differences in the existence of molecular regulators (*lsy-6*) and molecular effectors (*gcy* genes) of *C. elegans* ASE laterality suggest that the ASE neurons of *P. pacificus* may not be functionally lateralized.

## Patterns of neuronal dye-filling are largely conserved

To further explore homologous features of *P. pacificus* and *C. elegans* amphid neurons, we visualized the organization and location of the neuronal cell bodies and their dendritic processes in live wild-type animals with the lypophilic dyes DiI and DiO. In *C. elegans* non-dauer hermaphrodites, DiI, DiO, and FITC routinely stain five specific pairs of amphid neurons with open sensory endings to the environment - ASK, ADL, ASI, ASH, ASJ (*Table 3*; *Figure 4B and I*) - along with two pairs of tail phasmid sensory neurons - PHA and PHB. Additionally, ADF is stained weakly by FITC but not by DiI, while AWB is stained weakly by DiI but more strongly by DiO (*Hedgecock et al., 1985*; *Perkins et al., 1986*; *Starich et al., 1995*). It is unclear why only certain amphid neurons exposed to the environment take up certain dyes. In *C. elegans*, the cell bodies of the three dorsal-most amphid neurons just below the amphid sheath cell form a distinctive DiI-stained trio (ASK, ADL, ASI), while ASH and ASJ are visible at different focal planes ventral or posterior to this trio, respectively (*Figure 4*). The cell ablations of the DiI positive ASH neurons show that its polymodal function is strongly

**Table 3.** Dye filling properties of individual amphid neurons in young adult hermaphrodites.
orange = clearly conserved; blue = clearly divergent; in brackets()=weak staining, *weak FITC uptake in *C. elegans*.

| Amphid Homologs | | *P. pacificus* | | *C. elegans* | |
|---|---|---|---|---|---|
| *Ppa* | *Cel* | DiI | DiO | DiI | DiO |
| AM1 | ASH | + | - | + | + |
| AM2 | ADL | + | + | + | + |
| AM4 | ASK | + | - | + | + |
| AM8 | ASJ | + | + | + | + |
| AM9 | ADF | + | - | -* | - |
| AM10 | ASI | - | - | + | + |
| AM11 | AWB | (+) | - | (+) | + |

DOI: https://doi.org/10.7554/eLife.47155.040

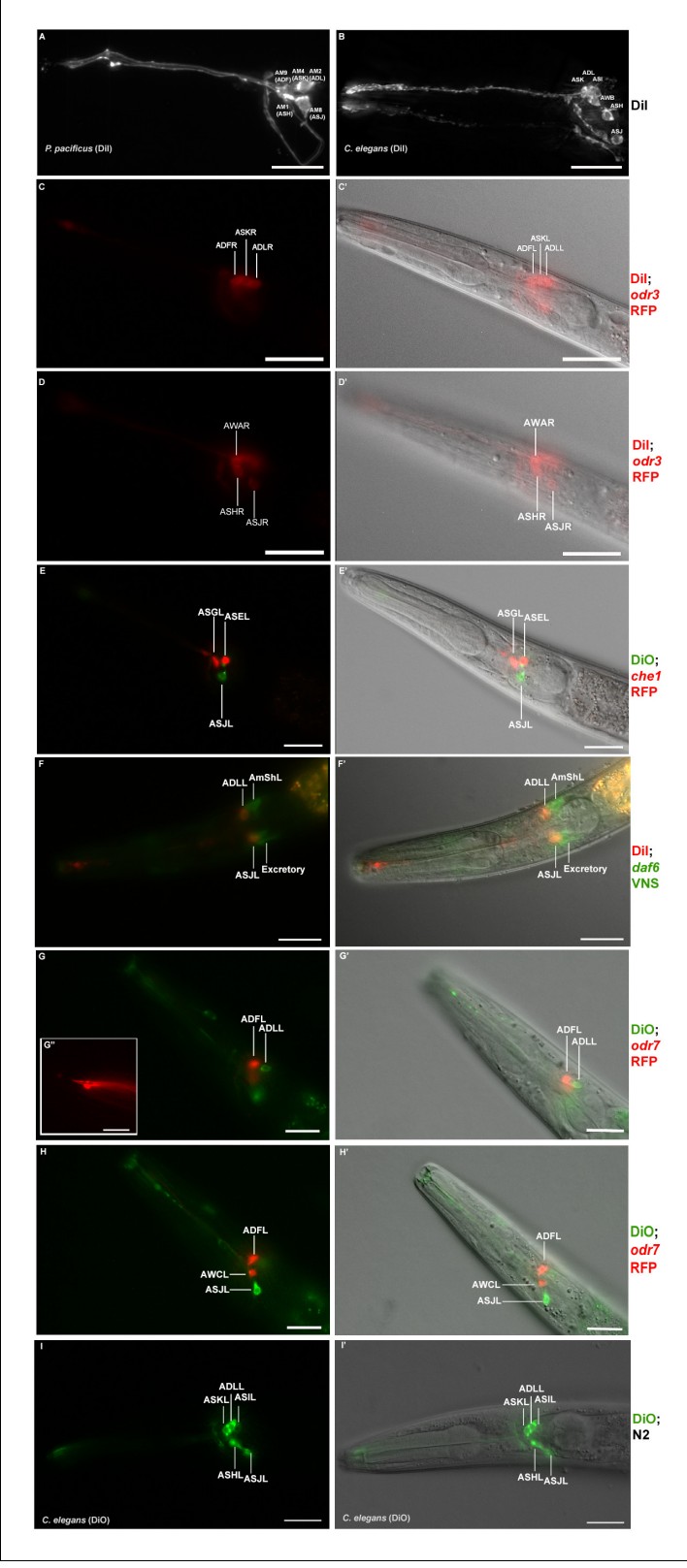

**Figure 4.** Dye filling and reporter gene expression in *P. pacificus* amphid neurons. (**A**) Stacked and deconvoluted fluorescent images of DiI stained amphid neurons in a *P. pacificus* young adult hermaphrodite (left lateral view). AM9(ADF), AM4(ASK), AM2(ADL), AM1(ASH), and AM8(ASJ) amphid neurons stain robustly. (**B**) In *C. elegans*, DiI stains the ASK, ADL, ASI, AWB, ASH, and ASJ neurons. (**C–H**) Single plane fluorescent images in *P. pacificus*. (**C–**

*Figure 4 continued on next page*

*Figure 4 continued*

D) DiI stained *Ppa-odr-3p::rfp* transgenic J3 larvae showing the three dorsal amphid neurons and the larger ASH and ASJ neurons in two focal planes. (E) *Ppa-che-1p::rfp* transgenic J4 hermaphrodite showing expression in ASE and ASG. (F) A DiI stained *Ppa-daf-6p::venus* J4 larva showing ADL and ASJ staining, anterior to the amphid sheath and the excretory cells, respectively. (G, H) A DiO stained *Ppa-odr-7p::rfp* young adult in two focal planes showing dye filling in ADL and ASJ, and RFP expression in ADF and AWC. (G' inset) The dendritic ends of another *Ppa-odr-7p::rfp* adult show the prominent PCMC of ADF with double cilia (bottom), and the smaller PCMC of ASK with a single cilium. (I) A DiO stained *C. elegans* young adult showing five dye filled neurons ASK, ADL, ASI, ASH, and ASJ. Scale bar: 5 µm. (C'–I') DIC overlay of the same DiI fluorescence images. Anterior is left and dorsal is up. Scale bar: 20 µm. (Representative images based on: *Ppa-odr-3p::rfp* n = 22; *Ppa-che-1p::rfp* n = 31; *Ppa-daf-6p:: venus* n = 60; PS312 n = 10; *Ppa-odr-7p::venus* n = 26). Additional Z-stacks are available: *Figure 4—figure supplements 1–5* and *Figure 1—videos 1–3*.

DOI: https://doi.org/10.7554/eLife.47155.031

The following video and figure supplements are available for figure 4:

**Figure supplement 1.** Receptor-type guanylyl cyclases in *P. pacificus* (red font) and *Caenorhabditis* species (black font).

DOI: https://doi.org/10.7554/eLife.47155.032

**Figure supplement 2.** Abbreviated phylogeny of ASE taste receptor-type guanylyl cyclases in *P. pacificus* (red font) and *Caenorhabditis* species (black font).

DOI: https://doi.org/10.7554/eLife.47155.033

**Figure supplement 3.** *Ppa-odr-7p::Cel-oig-8* mis-expression.

DOI: https://doi.org/10.7554/eLife.47155.034

**Figure supplement 4.** Orthology assignments for *che-1*, *odr-7* and *odr-3*.

DOI: https://doi.org/10.7554/eLife.47155.035

**Figure supplement 5.** Single molecule FISH of *Ppa-che-1*.

DOI: https://doi.org/10.7554/eLife.47155.036

**Figure 4—video 1.** Z-stack images of DiI stained J3 larva (ventral view).

DOI: https://doi.org/10.7554/eLife.47155.037

**Figure 4—video 2.** Z-stack images of *Ppa-che-1p::rfp* stained with DiO.

DOI: https://doi.org/10.7554/eLife.47155.038

**Figure 4—video 3.** Z-stack images of *Ppa-odr-7p::rfp*.

DOI: https://doi.org/10.7554/eLife.47155.039

conserved across various nematode species (*Srinivasan et al., 2008*). While ciliated channel neurons take up DiI differentially but are superficially conserved in diverse free-living nematode species examined, including other *Caenorhabditis* species, *Panagrellus redivivus*, and *P. pacificus*, DiI uptake patterns are less similar in the insect parasites such as *Steinernema carpocapse* and *Heterorhabditis bacteriophora*. Thus, DiI uptake is an important but not singular criterion for defining homologous sensory neurons (*Han et al., 2016*; *Srinivasan et al., 2008*).

In *P. pacificus* young adult hermaphrodites, DiI stains five pairs of amphid neurons in a pattern similar to the one in *C. elegans* [AM9, AM4, AM2(ADL), AM1, and AM8] (n > 50; *Figure 4A–B*) but DiO stains only the AM2(ADL) and AM8 neurons (*Figures 4E, G and H*; *Table 3*). In contrast to *C. elegans*, the *P. pacificus* phasmid neurons only dye fill in the dauer larvae (data not shown), while the AM11(AWB) neurons rarely dye fill in any developmental stage with either dye (AWB is stained more robustly with DiO than with DiI in *C. elegans*). The similarity in cell body positions for AM8 and morphology for both AM2 as well as AM8, suggest they are likely ADL and ASJ homologs, respectively. As mentioned earlier, AM2 axons are also the only neurons that resemble ADL for not passing through the commissure and branching in the dorsal-ventral direction, so that the AS2(ADL) homology can be considered well supported. Thus, DiI staining in the three dorsal-most cells in the same focal plane as the AMsh cell body has only superficial resemblance to the three dorsal-most neurons in *C. elegans* (*Figure 4C*; *Figure 4—video 1*). If the posterior AM2 cell in the trio is the ADL homolog, rather than the ASI as in *C. elegans*, then the middle cell of this trio, AM4, is likely the ASK homolog, while the anterior cell of this trio, AM9, is the ADF. In addition to the conservation in DiI staining, we have nominated AM9 as the ADF homolog primarily due to the conservation of a distinctively bulbous PCMC in the ultrastructure of the *P. pacificus* AM9 similar to the *C. elegans* ADF (*Figure 1G–H*) (*Doroquez et al., 2014*). Thus we assume that AM9(ADF), AM4(ASK), and AM2(ADL)

together make up the dorsal trio of neurons that take up DiI. In this proposed homology, the puta-tive ASI homolog, AM10, does not take up DiI. In *C. elegans*, the ASI neurons are important for reg-ulating dauer development and are remodeled in the dauer larvae such that the dauer ASI neurons no longer take up DiI due to cilia retraction from the amphid pore (*Albert and Riddle, 1983*; *Peckol et al., 2001*). Given that in *P. pacificus* the phasmid neurons only take up DiI as dauer larvae, it is possible that differential DiI uptake in homologous neurons between the two species recapitu-lates certain remodeling events in amphid neurons during dauer entry and exit that is lost in one lineage. Our cell homology nominations are consistent with the ASH and ASJ identified by position and DiI staining in a previous study by *Srinivasan et al. (2008)*, but are different for the DiI stained dorsal trio (ADF, ASK, ADL). *Table 3* summarizes the dye filling properties of the individual amphid neurons. While the positioning of the neurons taking up DiI appears mostly unaltered at first glance, cell physiology or the chemical environment for the live dye may have diverged modestly during evolution. An overview of the reconstructed cilia in one-to-one comparisons to their *C. elegans* coun-terparts is shown in *Figure 5*.

## The amphid sheath glia

To complement the characterization of the neuronal composition of the amphid sensillum, we set out to visualize the morphology of the glia-like amphid sheath cells (AMsh) in more detail and in vivo. To this end, we constructed an AMsh reporter using a 2.4 kb region of the *P. pacificus daf-6* promoter to drive the Red Fluorescent Protein (RFP). DAF-6 is a Patch-related protein required for proper tubule formation in *C. elegans*, including the morphogenesis of the amphid sheath channel (*Oikonomou et al., 2011*). Indeed, *P. pacificus daf-6* expression shows strong conservation not only in the AMsh, but also in cells of the excretory duct and pore, seam cells, as well as in the VulE epi-dermal cell that contributes to vulva formation during the mid-J4 larval stage (*Perens and Shaham, 2005*) (*Figure 6A*; *Figure 6—video 1*; data not shown). We found a similar *daf-6* expression profile in dauer larvae (*Figure 6B*). The cell bodies of the amphid sheath cells are distinctively large and sit dorso-anteriorly to the terminal bulb of the pharynx. The AMsh cells extend thick processes anteri-orly that swell to form vesicle-filled paddle-like endings near the nose of the worm (*Figure 2A–B*,

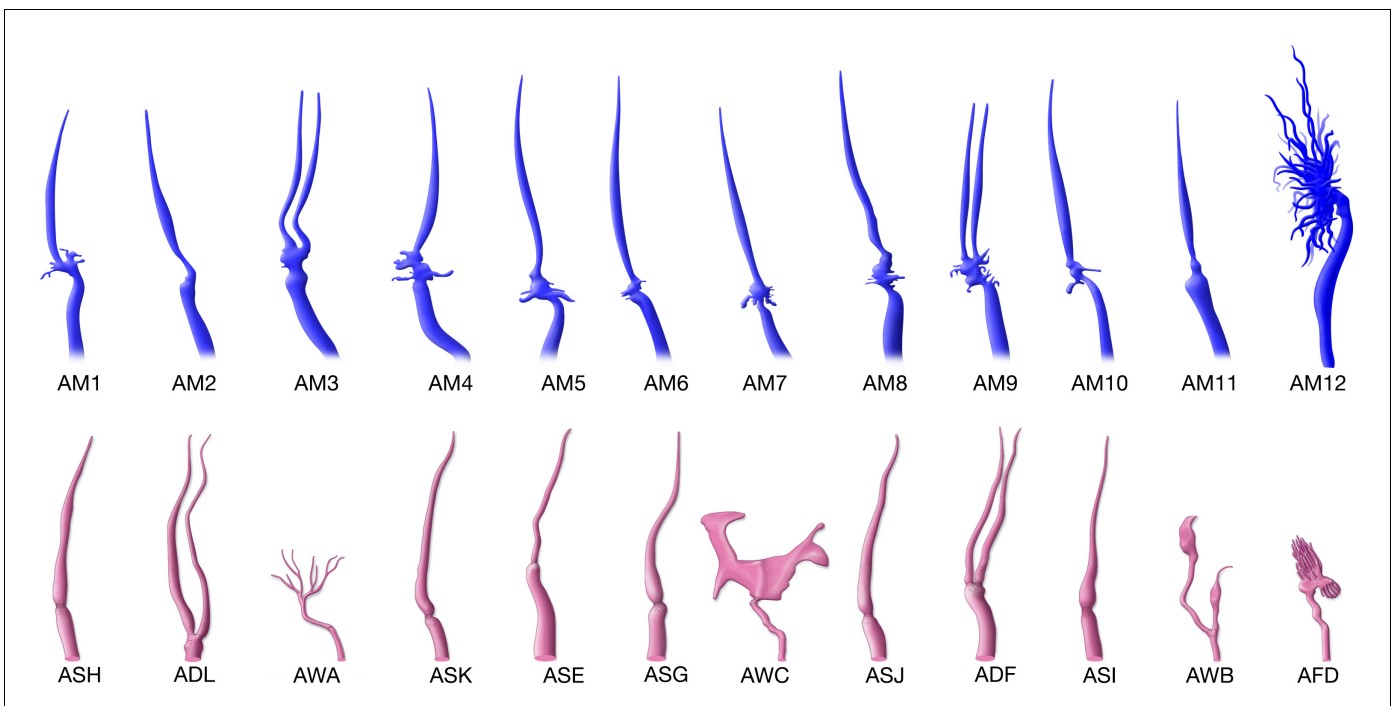

**Figure 5.** Comparison of the reconstructed cilia of *P. pacificus* and *C. elegans* amphid neurons. *P. pacificus* lacks neurons with wing-like morphology.
DOI: https://doi.org/10.7554/eLife.47155.041

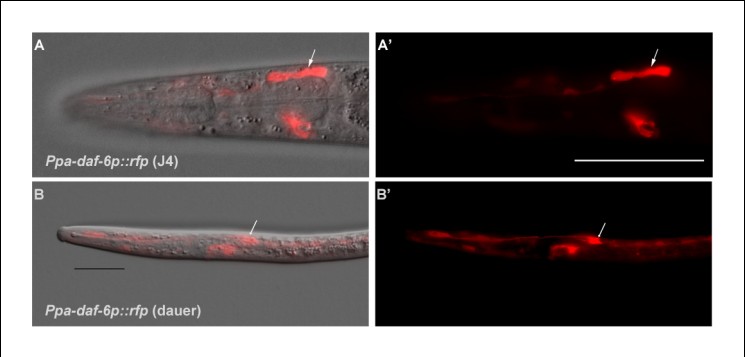

**Figure 6.** Genetic marker for the amphid glia. (**A–A'**) DIC overlay and *Ppa-daf-6p::rfp* expression in the dorsally located amphid sheath cell as well as the excretory cell on the ventral side of a J4 hermaphrodite. Scale bar: 50 μm. (**B–B'**) DIC overlay and *Ppa-daf-6p::rfp* expression in a dauer larva with prominent amphid sheath, excretory cell, and seam cells expression. Scale bar: 25 μm. Arrows indicate the amphid sheath cell body. A Z-stack of a *Ppa-daf-6p::rfp* adult is available: *Figure 6—video 1*.
DOI: https://doi.org/10.7554/eLife.47155.042
The following video is available for figure 6:
**Figure 6—video 1.** 3D rendering of *Ppa-daf-6::rfp* J4 with confocal microscopy.
DOI: https://doi.org/10.7554/eLife.47155.043

*Figure 3A–B*), unlike their *C. elegans* counterparts, whose endings fan out widely into the head, forming large sheets to accommodate the expanded ciliated endings of the winged AWC neurons and the finger cells. The amphid socket cells distally surround the ciliated ends of the sensory dendrites, each forming autocellular junctions onto themselves to create a pore in the cuticle of the lateral lip that is in direct contact with the environment (*Figure 1C–D*; *Figure 2A–B*). The sensory dendrites enter the matrix-filled lumen of the amphid sheath cell process quite distal to their respective cell bodies. The lumen, or channel, is an extracellular space formed by the merging of numerous large matrix-filled vesicles (*Figure 1G–H*; *Figure 2A–B*). Accordingly, the cytoplasm of the posterior process and the cell body is rich in mitochondria, ribosomes, rough endoplasmic reticulum, Golgi complexes and vesicles of different sizes, distinguishing the AMsh as an active secretory cell (*Figure 1I–K*). We speculate that some of the secreted factors act as accessory to chemosensory function to modify odors or water soluble molecules (*Bacaj et al., 2008*; *Cinkornpumin et al., 2014*), or to protect the neurons against reactive oxygen species (*Liu et al., 2017*; *Liu et al., 2015*).

In contrast to *C. elegans*, all the non-finger amphid neurons in *P. pacificus* have dendritic processes that terminate in the amphid channel and are thus in direct contact to the external environment (*Figure 1A–F*, *Figure 2A–B*). Because the *C. elegans* AMsh cells accommodate the prominently large AWC winged neurons, the left and the right AMsh can fuse to each other in the dauer larvae, which undergo radial constriction during dauer entry (*Procko et al., 2011*). Given the absence of any amphid neurons with winged morphology in *P. pacificus*, not surprisingly, *Ppa-daf-6p::rfp* expression in the pair of anterior AMsh endings remains distinctively separated in *P. pacificus* dauer larvae (*Figure 6B*). While the *C. elegans* amphid sheath processes expand into a large sheet at their anterior ends to accommodate the wing-shaped cilia, the amphid sheath processes in *P. pacificus* maintain a tube-like form, becoming slightly wider in diameter towards the anterior tip, where they take up the 12 neurons, and thinner posteriorly near the nerve ring in an apparent kink anterior to the cell body (n = 3). As a result, the *P. pacificus* amphid sheath morphology is tubular, lacking the distal enlargement of the *C. elegans* amphid sheath.

## Dendrite entry into the sheath cell
Next, we examined the interaction of the neuronal dendrites with sheath cells in more detail. During *C. elegans* development, the nascent dendrites of the amphid neurons and the processes of the sheath glial cells attach to the tip of the nose and from there elongate posteriorly by retrograde growth when the cell bodies start migrating posteriorly to their final position. This elongation process happens in a precisely coordinated way in each bundle (*Heiman and Shaham, 2009*).

Reconstruction of transverse TEM sections from four *C. elegans* specimens show invariant arrangement of the amphid neurons in the sensory channel, such that each amphid neuron can be reliably identified by its position in the channel (*Ward et al., 1975*). However, recent studies using live imaging on a much larger number of *C. elegans* samples reveal some variability in dendrite entry during larval development (*Yip and Heiman, 2018*). In our data set of only two specimens, we saw a variation in the channel position of one cilium even between the left and right side in one animal. To determine the degree in which the order and position of each dendrite entry is stereotypical in *P. pacificus*, we determined the individual entry points of amphid dendrites into the sheath in the reference sample (Specimen 107, similar data from specimen 148 not shown; *Figure 2—figure supplements 1* and *2*). Over most of their length the dendrite bundles are located on the ventral side of the AMsh but from 36.4 μm and 39.5 μm from the tip of the head on the left and right sides, respectively, shortly before the tips of the AMsh processes start to become wider, the bundles loosen and dendrites distribute around the sheath processes, finally arranging themselves into a dorsal and a ventral group. The first dendrites to enter are those of the finger neurons AM12(AFD) at approximately 25 μm from the tip of the head, or 1/6 the remaining distance between the posterior end of the pharynx (section 3000) and the tip of the 'nose' at the anterior end (Section 0). Anterior to the region occupied by AM12(AFD), the dendritic bundle splits into two or three groups consisting of one or more dendrites that enter the sheath in variable order from two or three sides. On the left side, for example, AM5(ASE, orange), AM4(ASK, bright green), AM6(ASG, light blue) and AM3 (AWA, cream) form a group which enter laterally whereas on the right side there is no lateral group. Instead, AM5(ASE, orange) and AM4(ASK, bright green) enter from the ventro-lateral side, while AM6(ASG,light blue) and AM3(AWA, cream) are part of the dorsal entry group. The entry of the finger neuron AM12(AFD, yellow) is also variable, it can be dorsal as on the left side or ventral as on the right side. In contrast, the site of entry for AM8(ASJ, middle blue) is always ventral and for the remaining seven other neurons consistently dorsal on both sides. The dendrites of AM11(AWB, magenta red) are the last to enter the sheath (dorsally) 13.5 and 13.8 μm from the tip of the head. Thus, the sequence and site of entry for many neurons relative to the amphid sheath are variable in *P. pacificus*, but higher sample size involving semi-automated live imaging would be necessary to determine if the left-right variability is consistent and if the degree of variability is different between *P. pacificus* and *C. elegans*.

## Homologies of other sensory neurons

In addition to the sensory sensilla, both *P. pacificus* and *C. elegans* also possess five types of sensory receptors that terminate in the nose region of the animal but are not accompanied by sheath or socket cells. Of these, the BAG neurons (named for bag-like sensory ending) stand out for being required for $CO_2$ and $O_2$ sensing in *C. elegans* (*Hallem and Sternberg, 2008*; *Zimmer et al., 2009*), as well as $CO_2$ avoidance in *P. pacificus* (*Hallem et al., 2011*), both of which are presumably important for host detection in parasitic nematodes. In *P. pacificus,* the two BAG neuron cell bodies are located just anterior of the nerve ring in subventral position close to the isthmus (*Figure 3A–B*, *Figure 7A–B*), with their dendritic processes running in the ventral-most position of the amphid process and lateral labial bundle. The ciliated distal ends of these processes form branched lamellae around the ventral halves of the lateral inner labial socket cells (ILsoL), opposite to those of another pair of internal receptors, the URX neurons (*Figure 1L*, *Figure 7A–B*, *Figure 1—figure supplement 1* and *Figure 1—video 1*). In *C. elegans,* the pair of URX neurons is known to be important for sensing oxygen levels and to control carbon dioxide response (*Carrillo et al., 2013*; *Gray et al., 2004*). The cell body of the URX neuron in both nematode species is located posterior to the nerve ring, and directly anterior to the anterior-most amphid neuron cell body in dorsal position [AM9(ADF) in *P. pacificus* and ASK in *C. elegans*] (*Figure 3B*). Interestingly, the dendritic endings of the homologous pair of *P. pacificus* URX neurons appear to be ciliated with eight microtubules, which are associated with the dorsal half of the lateral IL socket cells (*Figure 1L*, *Figure 7A–B*). In contrast, the *C. elegans* URX neurons are non-ciliated and unaffiliated with any sensilla (*Doroquez et al., 2014*). Ciliated URX neurons are also found in other nematode species, for example, in one of two dendritic endings of the URX neurons in the soil-dwelling nematode *Acrobeles complexus*, as well as in all of the URX endings of the mycophagus nematode *Aphelenchus avenae* (*Bumbarger et al., 2007b*; *Ragsdale et al., 2009*). Similarly, the four putative URY neurons have dendritic endings with numerous (6-20) singlet microtubules but they do not appear to be ciliated. These extensions are split into

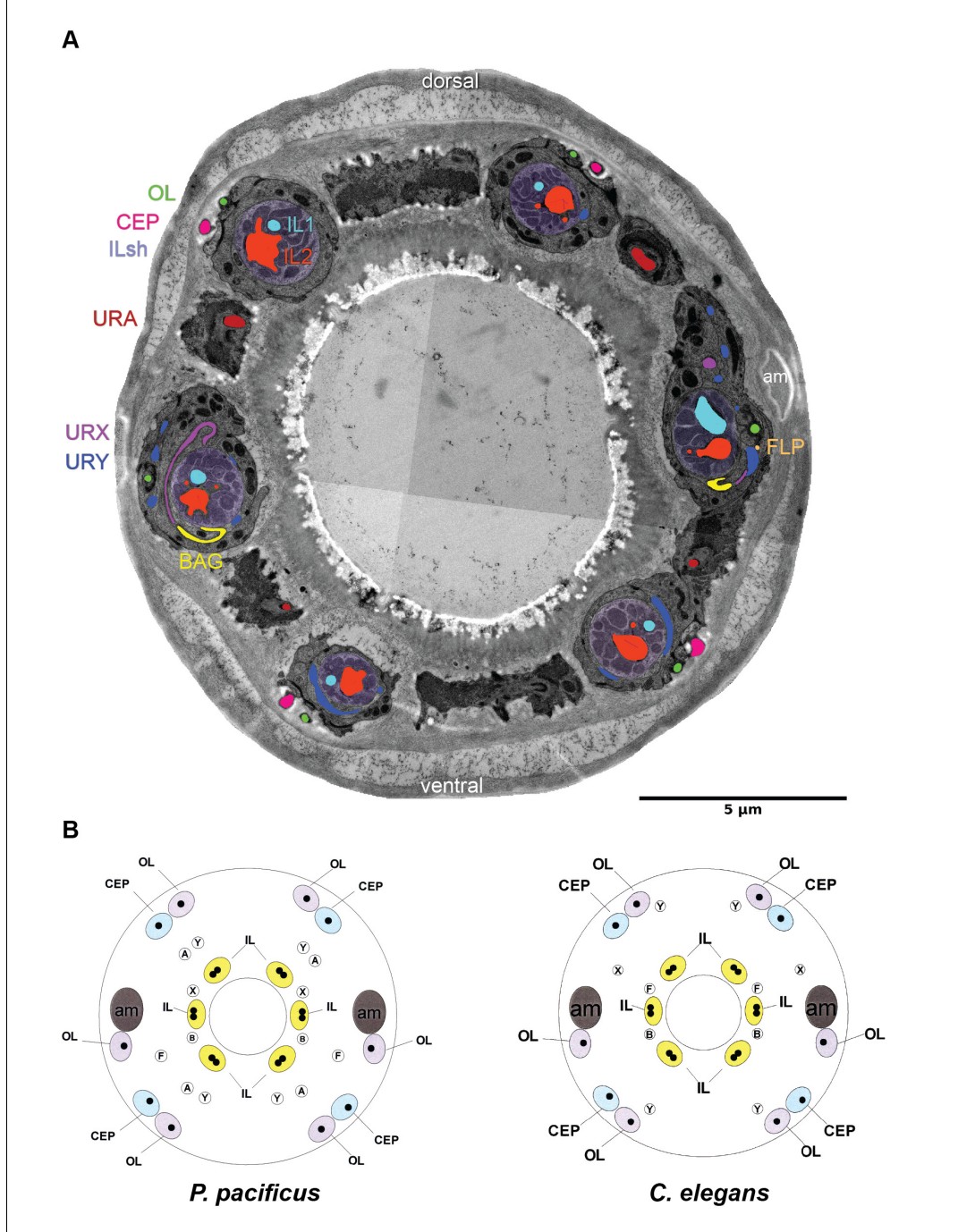

**Figure 7.** Comparison of cuticular sensilla and internal receptors. (**A**) Transversal TEM section of *P. pacificus* cuticular sensilla and internal receptors of the mouth region. One from each sensillum type is labeled in the same false color as the neuron. Note the highly branched dendritic endings of the four URY neurons (blue), which associate with all six inner labial sensilla forming sheet-like branches around the processes of the ILsh and ILso cells. (**B–C**) Schematic illustrations of the cuticular sensilla in *P. pacificus* and *C. elegans*. Inner Labial 1 and 2 (IL), Inner Labial sheath (ILsh), Outer Labial (OL), Cephalic (CEP), URA (**A**), URY (**Y**), URX (**X**), BAG (**B**), FLP (**F**), amphid (am).

DOI: https://doi.org/10.7554/eLife.47155.044

several overlapping branches extending membranous elaborations towards all six ILsh and ILso, resembling their *C. elegans* counterparts (*Figure 1—figure supplement 2* and *Figure 1—video 2*) (*Doroquez et al., 2014*). Lastly, the four putative URA neurons of unknown function also resemble the morphology of their homologs in *C. elegans* (*Figure 1—video 3*). A comparison of *C. elegans*,

*P. pacificus*, and *A. complexus* suggests that wing neurons (AWx) and unciliated URX neurons are derived characters in *C. elegans*, whereas ciliated BAG and unciliated URY are conserved (*Figure 1—figure supplement 3*).

The *C. elegans* FLP neurons have elaborate multi-dendritic structures throughout the head region, and are polymodal receptors for thermosensation and mechanosensation (*Albeg et al., 2011*; *Chatzigeorgiou and Schafer, 2011*). Like in *C. elegans*, the cell bodies of the likely FLP homologs in *P. pacificus* are located posterior to all of the amphid neuron cell bodies near the terminal pharyngeal bulb and have dendrites with extensive branching, whereas their axon projections do not enter the nerve ring. Unlike the FLP neurons in *C. elegans* however, the *P. pacificus* FLP neurons do not appear to be associated with the lateral IL socket cell (*Doroquez et al., 2014*), such that the most anterior process we traced on the left side of Specimen 107 terminates in close proximity to URX and BAG in a region where the BAG neurons start to form their lamellae around the IL socket (*Figure 7A–B*). In *C. elegans,* the BAG and FLP neurons are unusual among the anterior sensory neurons for not possessing their own set of glial cells, but instead associate with the socket cells of the lateral IL neurons (*Doroquez et al., 2014*). The socket cells could interact with any nearby neuron or provide functional support, since it is not clear if the *C. elegans* BAG and FLP association with the IL socket cells is primarily for structural stability. It also remains to be determined what functional significance is there, if any, for *P. pacificus* URX neurons rather than the FLP neurons to associate with the lateral IL sensilla.

Another unusual neuron pair, called AUA in *C. elegans* (for Amphid Unknown Type A), has a clear homolog in *P. pacificus*. Although not a true amphid sensory neuron, the *C. elegans* AUA also sends its axon through the amphid commissure and possesses a distinctive dendrite-like process (*White et al., 1986*). While the axons of the AUA neuron pair are similar to many other amphid sensory neurons, also terminating by making a gap junction between the left and right partners at the dorsal midline, their dendrites terminate just anterior of the nerve ring rather than projecting to the anterior end of the animal. We observed a similar neuron in *P. pacificus,* AMU1 (AMphid Unknown 1), whose dendritic-like process terminates just anterior of the nerve ring (*Figure 3A–B*). Given this strong dendritic structural similarity, as well as a similar axonal projection, AMU1 is the likely homolog of AUA. The conservation of the AMU1(AUA) neurons, which mediate oxygen sensing and social feeding behavior in *C. elegans*, implies that they share conserved functions in several nematode species (*Bumbarger et al., 2009*; *Chang et al., 2006*; *Coates and de Bono, 2002*).

## Axonal process neighborhood is highly conserved

We next explored amphid neuron axon process placement within the nerve ring. The original characterization of the *C. elegans* nervous system identified and used reproducible placement of axons within ganglia as criteria to identify individual neurons (*Ware et al., 1975*; *White et al., 1986*). We used a similar approach and compared axonal placement in *P. pacificus* to *C. elegans* (*Figure 8A*). Specifically, we evaluated the ultrastructure of the ventral ganglion (*Figure 8B and D*), which is proximal to where the amphid commissure enters the main axonal neuropil, as well as the dorsal midline (*Figure 8C and E*), where most amphid axons terminate. Using the *P. pacificus* color codes, we labeled the location of each amphid axon in both species. Despite differences in sample preparation (HPF for *P. pacificus* and chemical fixation for *C. elegans*) and EM section thickness (50 nm for *P. pacificus* and ~80 nm for *C. elegans)*, we observed striking similarities in axonal placement in both species, providing further strong support for our homology assignment of neurons. For example, we observed similar amphid sensory neuron axonal fasciculation along the ventrodorsal and mediolateral axes of the ventral ganglion, where the ASJ and ASG neurons are most lateral, and the AFD and AUA neurons are the most ventral in both species (*Figure 8B and D*). The most posterior segment of the dorsal nerve cord, where most bilaterally symmetric neurons meet, is occupied by a similar set of neurons in both species (*Figure 8C and E*). Also, of note are the ASJ neurons, which are most ventral, and ADL, which are most lateral in both *P. pacificus* and *C. elegans*.

## Soma and process morphology of amphid interneurons are conserved

We then examined the next processing layer of amphid sensory information in the worm, the first-layer amphid interneurons. In *C. elegans* there are four such amphid interneuron classes, each composed of a bilaterally symmetric pair of neurons (termed AIA, AIB, AIY, AIZ, for Amphid Interneuron

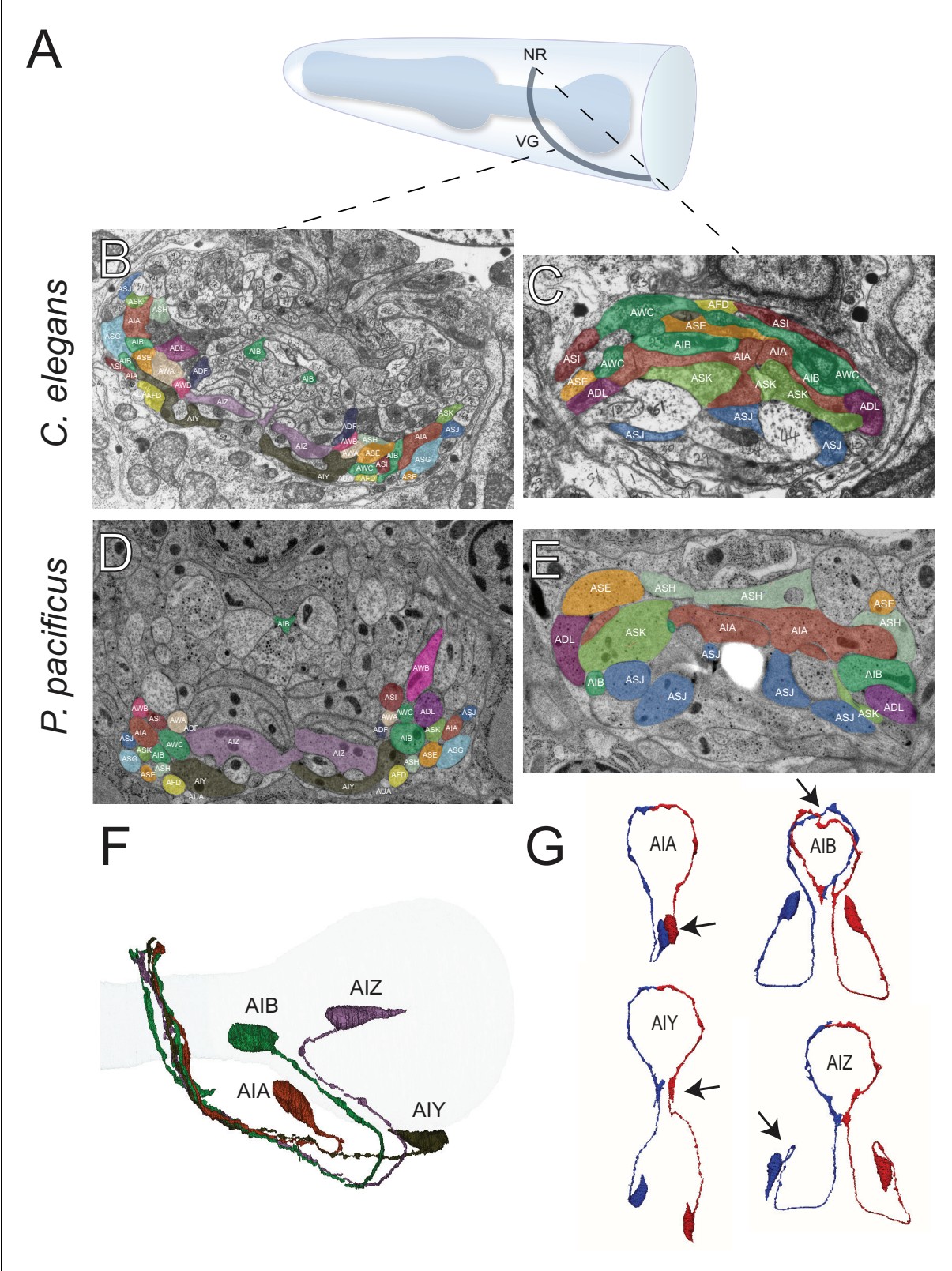

**Figure 8.** The first layer amphid interneurons. (**A**) Model of the anterior *P. pacificus* highlighting the nerve ring (NR) and ventral ganglion (VG). (**B**) Electron micrograph of the anterior *C. elegans* (N2U sample) ventral ganglion and dorsal nerve ring (**C**) colored with the same color code as in *Figure 1*. Also shown are AIA (maroon), AIB (dark green), AIY (green/gray), and AIZ (purple). *C. elegans* neuronal nomenclature is used. Electron micrograph of the anterior *P. pacificus* (specimen 107) ventral ganglion (**D**) and dorsal nerve ring (**E**) colored with the same color code as in *Figure 1*. (**F**) *Figure 8 continued on next page*

*Figure 8 continued*

Three dimensional rendering of the AIA, AIB, AIY, and AIZ interneurons from *P. pacificus* (specimen 107) with pharyngeal outline shown in light teal. (G) Individual three dimensional renderings of *P. pacificus* (specimen 107) amphid interneurons shown with a dorsoposterior view with the left and right neurons in blue and red, respectively. Arrows show anatomical features present in both species: AIA cell bodies are adjacent, AIB axons undergo a dorsal neighborhood change within in the nerve ring, AIY neurons have varicosities relating to synaptic output posterior to the nerve ring, and AIZ axons show anterior projections before commissural entry. Amphid neuroanatomy can be compared to *C. elegans* at http://www.wormatlas.org/images/NeuronImageList.jpg.

DOI: https://doi.org/10.7554/eLife.47155.045

A, B, Y and Z). All four neurons display highly distinctive features and an examination of the *P. pacificus* nervous system revealed a remarkable extent of conservation of these features, thereby easily identifying these neurons (the ease of identification prompted us to skip the tentative numerical naming scheme that we initially applied to sensory neurons, and thus we assigned these *P. pacificus* neurons with the same name as in *C. elegans* straight away). In both species, the AIY interneuron cell bodies are located in the same relative position in the ventral ganglion and their axons display a characteristic 'humped' morphology (*White et al., 1986*). This 'humped' morphology corresponds to a large synaptic output onto AIZ, which forms a sheet-like cross-section immediately dorsal to AIY (*Figure 8B and D*). The AIB neurons are identifiable in both species by a characteristic switch of their process into two distinct neighborhoods of the anterior portion of the ventral ganglion, where its proximal axon is ventral and its distal axon is more dorsal. The somas of the *P. pacificus* and *C. elegans* AIA interneurons occupy a stereotypic mediodorsal location and, like the AIY and AIZ neurons, form a dorsal midline gap junction. We rendered the complete structure of the amphid interneurons in 3D, further illustrating that cell body locations and axon projections are nearly identical in both species (*Figure 8F*).

## Patterns of conservation and divergence in synaptic connectivity

Having identified the first-layer amphid interneurons in *P. pacificus*, we next explored the degree to which synaptic connectivity in the amphid circuit is conserved across species. We annotated all chemical synapses and gap junctions between the *P. pacificus* amphid sensory neurons, AUA, AIA, AIB, AIY, and AIZ, recording both the number of individual synapses as well as the number of serial section electron micrographs where ultrastructural synaptic anatomy was present as a proxy for anatomical connection strength. We identified 138 chemical connections (directed edges in a graph of connectivity) and 98 gap junction connections (undirected edges) in *P. pacificus*, compared to 73 chemical edges and 96 gap junction edges in a recent re-evaluation of connectivity in *C. elegans* (*Cook et al., 2019*). As small synapses are more difficult to annotate reliably (*Xu et al., 2013*), we limited our analysis to only include connections that are ≥10 EM sections in strength in both species, which represents the strongest 50% of synaptic connections. Such thresholding also should minimize any potential concerns about comparing synaptic annotations between distinct datasets of different provenance.

Of these strong chemical synaptic connections, 32/53 connections were present in both species while 15 and 6 were specific to *P. pacificus* and *C. elegans*, respectively. Four of the strong gap junction connections were present in both species while 6 and 2 were specific to *P. pacificus* and *C. elegans*, respectively (*Figure 9A*). To better contextualize similarities and differences in connectivity, we created a circuit diagram that shows a layered output from sensory neurons (triangles) onto interneurons (hexagons). We found that, on average, conserved edges are larger and more frequently made by amphid interneurons. Moreover, neurons whose structure is qualitatively most similar between species made more similar synaptic outputs (*Figure 9B*). Examples include the synaptic output of AFD, whose output is almost exclusively onto AIY, or the patterns of interconnectivity among the amphid interneurons. In contrast, the amphid wing neurons (AWA, AWB, AWC) showed multiple connections present in only one species. For example, AWA makes strong synaptic connections to the AIB interneurons exclusively in *P. pacificus*. Similarly, the *P. pacificus* polymodal ASH neuron class makes *P. pacificus*-specific outputs to AIY and is innervated by AIA. Similarly, the AUA is presynaptic to the RIA and RIB interneurons in both species. Like in *C. elegans*, the AUA neuron is the exclusive target of the ADF neurons. While many of the *P. pacificus*-specific synaptic contacts concern the connection of sensory neurons to first layer interneurons, many of the *C. elegans*-

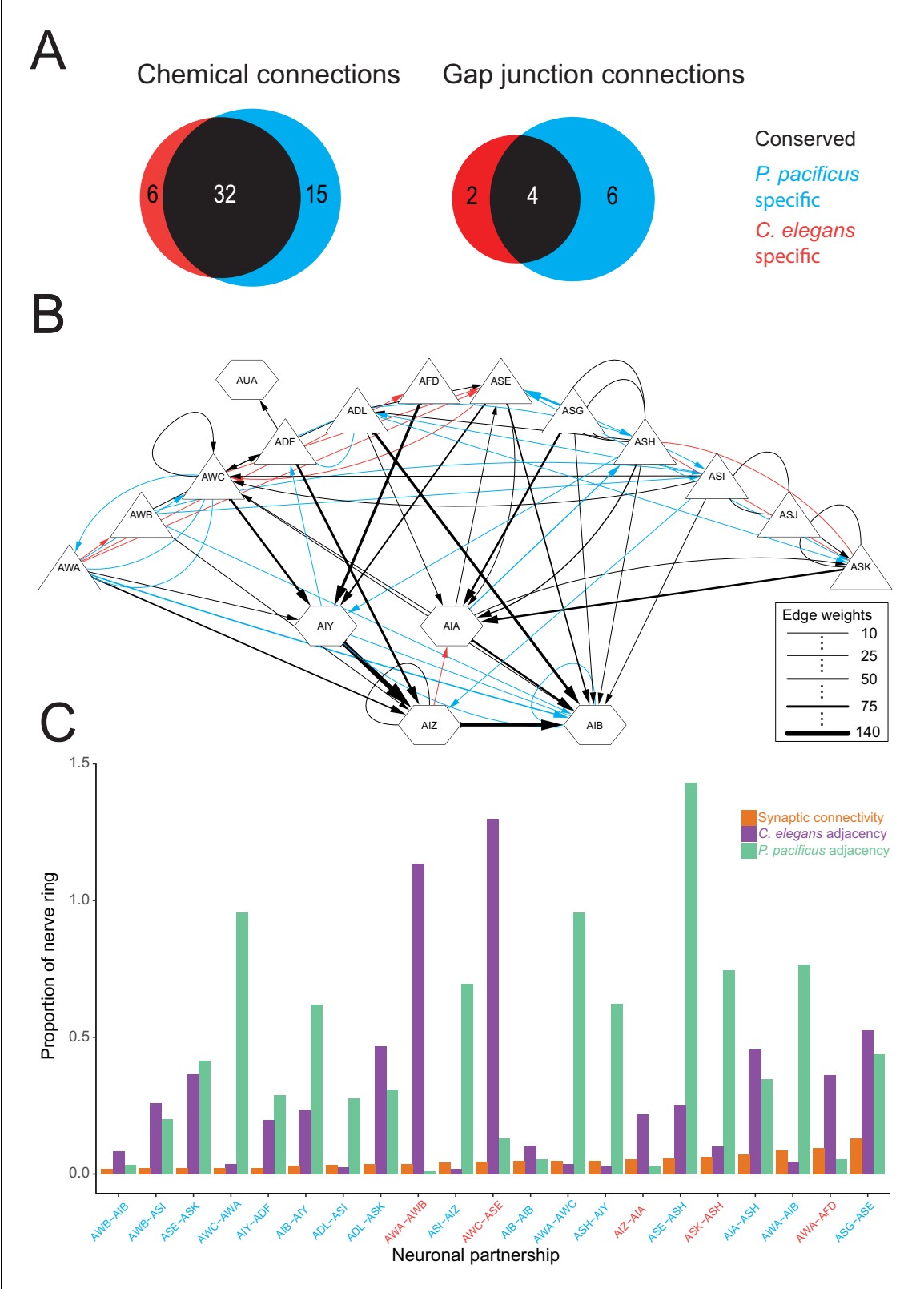

**Figure 9.** Comparisons of synaptic connectivity. (**A**) Venn diagrams showing number of conserved (black), *C. elegans*-specific (red), and *P. pacificus*-specific (blue) chemical connections (directed edges) and gap junction connections (undirected edges). Threshold for edge weight was set to ≥10 serial EM sections, connectivity was determined between amphid sensory neurons, AUA, and amphid interneurons AIA, AIB, AIY, and AIZ. (**B**) Circuit diagram showing sensory neurons (triangles) connecting to interneurons (hexagons) by chemical synaptic connectivity (arrows) and gap junctions (lines). The

*Figure 9 continued on next page*

*Figure 9 continued*

thickness of the line is proportional to the weight of connection (aggregate number of serial section electron micrographs where a synaptic specialization is observed, with only connections ≥ 10 sections shown in this diagram). (C) Comparison of species-specific connections and neuronal adjacency. Bar graph showing chemical synaptic connectivity (orange), neuronal adjacency for *C. elegans* (purple), and neuronal adjacency for *P. pacificus* (green). The x axis is ordered by increasing strength of species-specific connection, and colored coded as in 8A. Adjacency was determined computationally by comparing adjacent pixels of volumetrically traced neuron profiles. Proportion was determined by dividing the number of serial EM sections showing a synaptic connection or adjacency by the total number of sections comprising the nerve ring and ventral ganglion. A proportion of 1.0 indicates the entire ipsilateral region of interest was adjacent, with proportions over 1.0 showing adjacency on both sides of the nerve ring. Each partnership evaluated is a combination of two neurons that formed a species-specific connection shown in (B). *Figure 9—source datas 1–3*.

DOI: https://doi.org/10.7554/eLife.47155.046
The following source data is available for figure 9:

**Source data 1.** Species-specific connectivity of *P. pacificus* amphid neurons.
DOI: https://doi.org/10.7554/eLife.47155.047
**Source data 2.** Connectivity of *P. pacificus* amphid neurons.
DOI: https://doi.org/10.7554/eLife.47155.048
**Source data 3.** Connectivity of amphid neurons *P. pacificus* and *C. elegans*.
DOI: https://doi.org/10.7554/eLife.47155.049

specific synaptic contacts concern the connections between sensory neurons. This may indicate an increase in cross-communication of sensory modalities in *C. elegans* as well as distinctions in which sensory outputs are processed in *P. pacificus.*

Synaptic connections present in only one species could be due to either differences in the structure of neuronal neighborhoods or synaptic specificity among adjacent processes. To distinguish between these possibilities, we computationally determined all axon-axon adjacencies of the amphid circuit in both species as previously described (*Brittin et al., 2018*). To compare across species, and to reduce section thickness and fixation artifacts, we calculated the proportion of adjacency (total number of EM sections where two processes are adjacent divided by the number of sections in the region of interest), similar to previous analyses of neuronal adjacency (*White et al., 1983*). Of the 29 species-specifc connections, there are only 21 pairs of possible neuron-neuron adjacencies due to reciprocal connectivity. Of these 21 neuron-neuron pairs, we found that all make physical contact in both species. The amount of contact did, however, vary across a wide range in both species. Among species-specific neuronal partnerships, we found the percent of axonal-axonal contact ranged from 0.95% to 142% (median contact 26.7%) (*Figure 9C*). We evaluated whether the amount of neuron-neuron adjacency could predict the strength of a synaptic connection. We compared the correlation between synaptic connectivity and neuron-neuron adjacency and found a very weak, non-significant, correlation in both species (Spearman's R = 0.2042, p=0.3747 in *P. pacificus*; Spearman's R = 0.1248, p=0.5898 in *C. elegans*). Together, these results suggest that species-specific differences in connectivity are largely determined by synaptic partnership choice or recognition rather than large differences in axonal fasciculation and neuronal neighborhood changes.

## Similarity and divergences of molecular features

Moving beyond anatomy, we assessed the evolutionary divergence of molecular features of individual neurons. Divergences in molecular features between homologous neurons in different species can be expected to be manifested via the species-specific loss or gain of genes that control certain neuronal phenotypes or via the modifications of expression patterns of genes. There appears to be ample evidence for both. For example, several censuses taken for the number of GPCR-type sensory receptor-encoding genes indicated a much smaller number in *P. pacificus* compared to *C. elegans* (*Dieterich et al., 2008*; *Krishnan et al., 2014*). Similarly, there are species-specific gene losses and gains in the complement of specific subfamilies of taste receptor-type guanylyl cyclases, as discussed above (Figure 4- figure supplement 1 and 2). On the level of regulatory factors, we noted above the absence of the *lsy-6* locus outside the *Rhabditidae*. In regard to genes that control structural features to neurons, we considered a gene, *oig-8*, which was recently identified as controlling the morphological elaborations of the winged cilia of the AWA/B/C neurons in *C. elegans* (*Howell and Hobert, 2017*). *oig-8* encodes a single transmembrane, immunoglobulin domain protein that not encoded in the *P. pacificus* genome. To assess whether this gene loss is responsible for the lack of winged cilia

in *P. pacificus,* we mis-expressed *Cel-oig-8* under the *Ppa-odr-7* promoter (described below) along with the *Ppa-odr-7p::rfp* marker but found no difference in the terminal dendritic cilia of RFP positive neurons that were co-injected with the *Ppa-odr-7p::Cel-oig-8* transgene compared to those that were injected only with the *Ppa-odr-7p::*RFP marker (*Figure 4—figure supplement 3*). This result suggests the two *Ppa-odr-7*-expressing amphid neurons may lack other factors (*DiTirro et al., 2019*), such as factors involved in ciliary membrane morphogenesis in wing neurons, to realize the branching function of *Cel-oig-8* .

To assess potential divergences in the expression patterns of conserved genes, we considered three genes that encode regulatory and signaling factors. For the regulatory factors, we chose two transcription factors that are expressed exclusively in a single neuron class in *C. elegans*, *odr-7*, an orphan nuclear hormone receptor uniquely expressed in AWA (*Sengupta et al., 1994*) and *che-1*, a zinc-finger transcription factor uniquely expressed in the ASE neurons (*Tursun et al., 2009*). Both transcription factors control the differentiated state of the respective neuron class in *C. elegans* (*Etchberger et al., 2007*; *Sengupta et al., 1994*; *Uchida et al., 2003*) and 1–1 orthologs could be identified in *P. pacificus* by reciprocal best BLASTP hits as well as protein sequence phylogeny (*Figure 4—figure supplement 4*). We generated a reporter gene fusion for the *Ppa-che-1* locus and found that transgenic *P. pacificus* expressed the reporter in two rather than one neuronal pairs in the head region of the worm (*Figure 4E*, *Figure 4—video 2*). The same expression can be observed by smFISH against the endogenous locus (*Figure 4—figure supplement 5*). By cell positions, as well as by the axons meeting at the dorsal midline using fluorescence microscopy, we provisionally identified these neurons as the AM5(ASE) and, tentatively, the AM6(ASG) neurons. Expression in AM5(ASE) matches the expression observed in *C. elegans* and expression in AM6(ASG) is notable because the ASG neuron is also a water-soluble chemical taste receptor (*Bargmann and Horvitz, 1991*) that can compensate for loss of ASE function under hypoxic conditions (*Pocock and Hobert, 2010*). Since the AM5(ASE) neurons of *P. pacificus* may not be functionally lateralized and, therefore, may not be able to discriminate between cues that are sensed by ASEL and ASER in *C. elegans*, we speculate that *che-1* may endow AM6(ASG) neurons of *P. pacificus* with chemosensory abilities that allow for the discrimination of ASE- and ASG-sensed water-soluble cues.

The expression pattern of the *odr-7* transcription factor seems to have diverged more substantially between *P. pacificus* and *C. elegans.* Based on cell body position and axonal projections, a *Ppa-odr-7p::rfp* reporter is expressed in the proposed AM7(AWC) and AM9(ADF) neuron pairs (*Figure 4G–H*, *Figure 4—video 3*, *Figure 4—figure supplement 4*), but not in AM3(AWA), the exclusive expression site of *odr-7* in *C. elegans*. This divergence of expression is particularly notable if one considers that *C. elegans odr-7* contributes to the specification of the *C. elegans*–specific elaborations of the AWA cilia (*Howell and Hobert, 2017*). In contrast, *P. pacificus* AM3(AWA) does not express *odr-7* and its cilia do not display the winged elaborations, as discussed above. Since we have not confirmed endogenous *odr-7* expression with smFISH, unlike with *che-1*, it is conceivable that our reporter construct lacks relevant cis-regulatory elements.

In contrast to the reduction of *gcy* genes important for *C. elegans* ASE laterality, the *P. pacificus* genome contains almost all of the downstream G-protein subunit signaling proteins found to be encoded in the *C. elegans* genome (*Figure 4—figure supplement 4*). We therefore established a transgenic reporter strain using the promoter of *Ppa-odr-3*, a G-protein subunit homolog known to be expressed in a number of *C. elegans* sensory neurons, most strongly in AWC, weaker in AWB, and faintly in AWA, ADF, ASH neurons in *C. elegans* (*Roayaie et al., 1998*). We found consistent *Ppa-odr-3p::rfp* expression in AM3(AWA) and less robust expression in AM4(ASK) (*Figure 4C–D*). Hence, like *odr-7* expression, the conservation of cell-type specific expression is limited to the amphid neurons but not to the proposed cellular homologs when soma position and axon projection patterns are also considered.

## Discussion

In the absence of morphological characteristics such as the winged endings of AWx neurons in *C. elegans*, we proposed neuronal homologies between *P. pacificus* and *C. elegans* amphid neurons based on their relative cell body positions, axon projections, the manner of dendrite entry into the amphid sheath cell, the number of cilia in channel, dye filling properties, and the connections to first layer interneurons. Using specific features present only in one of the neuron types in both species,

we have high confidence in identifying the homologs of ASH, ADL, ASJ, and AFD. We have lower confidence for other neuronal counterparts, particularly AWA and ASI, which lack most of these discriminating features. However, by the process of elimination, we are nevertheless confident about the homology assignments for the remaining neurons. Taking the analysis of all 12 amphid neurons together, we found modest conservation in synaptic connectivity, as well as both conservation and divergence in orthologous transcriptional reporters, *Ppa-che-1* and *Ppa-odr-7*, respectively. Although we have weighted each feature type equally, it is not likely that the same number of genetic changes separates each difference, resulting in the limitation and uncertainty of individual homology assignments as it is often noticed in comparisons of distantly related species.

In the future, functional studies targeting the individual *P. pacificus* amphid neurons will provide the necessary evidence for determining their sensory modalities, such as the AM5(ASE) in salt chemotaxis, the AM12(AFD) in thermotaxis, and the AM3(AWA) and AM7(AWC) neurons in olfaction. While such functional studies are of unique importance to elucidate the neuronal properties of the *P. pacificus* amphid system, any functional investigation can never fully overcome uncertainties in structural homology assignments because 'form and function' represent one of the three crucial antitheses of comparative biology (*Rieppel, 1988*). Additionally, homology at different hierarchical levels of biological organization can be independent from each other (*Riedl, 1978*; *Raff, 1996*). We found little overlap between the olfactory profiles of *P. pacificus* and *C. elegans*, also because the cell-specific markers, such as GPCRs, are themselves fast evolving (*e.g. C. elegans str-2* as a marker for lateral asymmetry). Ultimately, one criterion that might help solving some of the existing homology uncertainties is cell lineage reconstruction. Given that the pattern of cell divisions is largely invariant during development within individual nematode species (*Schierenberg and Sommer, 2014*; *Memar et al., 2019*), the 1-to-1 homology assignments in the amphid neurons may be resolvable by cell lineage analysis in a way that is currently only known from mollusks (*Katz, 2016*).

The genomes of *C. elegans* and *P. pacificus* are remarkably distinct. Detailed recent analyses using phylotranscriptomics, Illumina, and single molecule sequencing revealed striking differences in gene content and genome organization, which allow to more accurately date their divergence to ~100 million year ago (*Prabh et al., 2018*; *Roedelsperger, 2018*; *Rödelsperger et al., 2017*). In light of this divergence, the extent of similarities of nervous system patterning is remarkable. Neuron number appears invariant, neuronal soma position is restricted, and perhaps most remarkable are the similarities in process outgrowth and relative process position. Relative neighborhoods of processes are retained, including a number of remarkably subtle aspects of process morphology, such as the highly unique and characteristic neighborhood change of the AIB processes or the humped axon morphology of AIY at a specific location. These findings strongly suggest the existence of constraints in patterning of the nervous system, particularly during the phase of axonal and dendritic outgrowth, that is relative placement of processes into specific neighborhoods. Such placement is a critical pre-requisite for proper synaptic targeting choice by constraining which possible targets any given neuron can innervate (*White, 1985*). In light of these constraints, it is enlightening to observe a number of striking differences in synaptic connectivity. These changes are likely to generate very distinct avenues of information flow. Whether such changes in synaptic connectivity produce different types of behavior will require further investigation, since fascinating work in mollusks has shown that distinct wiring patterns of homologous neurons in distinct nudibranch species can also produce similar behaviors (*Katz, 2016*). In any case, our results argue that alterations in *en passant* synaptic connectivity between adjacent neurons, rather than initial patterning of neuronal fascicles, are a key substrate of evolutionary change. This is in striking contrast to evolutionary novelties in the olfactory systems of insects, where species-specific differences in olfactory behavior are achieved (among other mechanisms) by differences in the axonal targeting properties of olfactory sensory and projection neurons (*Ramdya and Benton, 2010*). Our work rather demonstrates striking constraints in axonal patterning, appearing invariant between *C. elegans* and *P. pacificus*. These constraints are even more notable if one considers the vast differences in chemotaxis behavior observed in *C. elegans* and *P. pacificus* (*Hong and Sommer, 2006*). Perhaps the main drivers of these chemotaxis differences are changes in olfactory perception, that is sensory anatomy and sensory receptors, rather than the means by which chemosensory signals are processed.

A previous comparative analysis of the pharyngeal nervous system, composed of an isolated circuit of 20 neurons, also found that homologous neurons types display differences in synaptic connectivity between *C. elegans* and *P. pacificus* (*Bumbarger et al., 2013*). However, in contrast to our

present study, this past analysis has not taken process adjacency into account, thereby leaving it unclear whether pharyngeal synaptic connectivity differences are the result of distinct process placement or distinct synaptic target choice within invariant process neighborhoods. Currently unpublished adjacency analysis of the *C. elegans* pharyngeal connectome suggests that pharyngeal neurons synapse onto almost all neighboring processes (S.J.C., unpubl. data), which is in striking contrast to the somatic nervous system where synapses are made only onto a fraction of neighboring processes (*White, 1985*). Differences in pharyngeal synaptic connectivity between *C. elegans* and *P. pacificus* may therefore be driven by distinct local process placements, a mechanism different from the one described here for the amphid sensory circuit.

There are genomic differences between *P. pacificus* and *C. elegans* that display tantalizing correlates to some of the specific neuroanatomical diversities. The three main olfactory neurons of *C. elegans* display elaborated cilia morphology (winged cilia) where olfactory receptors are known to localize. The *C. elegans* genome encodes more than 1300 olfactory-type GPCRs (*Robertson, 2006*; *Troemel et al., 1995*), many known to be co-expressed in AWA, AWB and AWC (*Troemel et al., 1995*; *Vidal et al., 2018*). In contrast, the *P. pacificus* genome contains significantly fewer olfactory-type GPCRs than *C. elegans* (*Prabh et al., 2018*), coinciding with less morphological complexity of olfactory cilia. It is tempting to speculate that the expansion of the olfactory receptor repertoire in *C. elegans,* relative to *P. pacificus*, and the concomitant expansion of the morphological elaborations in *C. elegans* are functionally coordinated. Yet despite the smaller GPCR repertoire, *P. pacificus* exhibits an odor preference profile that has scant overlap with *C. elegans* (*Cinkornpumin et al., 2014*; *Hong and Sommer, 2006*), hence one priority for future studies is to identify the amphid neurons that express odor receptors in *P. pacificus*.

Other genome sequence changes reveal intriguing correlates to a specific synaptic alteration that we observed. Like many other neuron pairs, the AM5(ASE) neurons of *P. pacificus* are electrically coupled. In *C. elegans,* the left and right ASE neurons are not electrically coupled (even though the processes are in contact to each other), thereby allowing both sensory neurons to discriminate between distinct sensory cues sensed by the left and right ASE neurons (*Ortiz et al., 2009*; *Suzuki et al., 2008*). Concomitant with the lack of electrical coupling of ASEL and ASER, a specific subset of ASEL and ASER-expressed rGC-type receptor proteins have expanded in the *C. elegans* genome (*Ortiz et al., 2006*), thereby expanding the spectrum of chemosensory cues that can be differentially sensed by ASEL vs. ASER. The genetic mechanisms to express this expanded set of *C. elegans*-specific receptor protein subfamilies in the left versus right ASE protein is triggered by the *lsy-6* miRNA, the most upstream regulator of ASEL/R asymmetry (*Cochella and Hobert, 2012*; *Johnston and Hobert, 2003*). Remarkably, this miRNA is a *Caenorhabditis* genus-specific 'invention', that is it does not exist in *P. pacificus* (*Ahmed et al., 2013*). In accordance with earlier suggestions (*Etchberger et al., 2009*; *Ortiz et al., 2006*), these findings indicate that ASE laterality arose from a bilaterally symmetric ground state.

One obvious question for the future is about the functional consequences of the observed differences, especially with regard to the different life styles of *P. pacificus* compared to *C. elegans*. While *C. elegans* is often found in rotten fruits (*Félix and Duveau, 2012*) and in association with slugs (*Petersen et al., 2015*), *P. pacificus* was first described from soil samples (*Sommer et al., 1996*) and subsequently found in reliable associations with several beetle species around the world (*Herrmann et al., 2007*; *Koneru et al., 2016*; *Ragsdale, 2015*). Additionally, *P. pacificus* nematodes have been found in insect baits for entomopathogenic associations (*Campos-Herrera et al., 2019*) and other vegetal substrates (*Félix et al., 2018*). However, none of these associations allows conclusions about the functional relevance of the divergent morphologies. Instead, they might represent examples of drift. We speculate therefore that 'network systems drift' could result in the observed patterns, similar to the developmental systems drift observed in vulva development between *C. elegans* and *P. pacificus* (*Wang and Sommer, 2011*). However, much work at the interface between neurobiology and ecology is necessary to confirm such hypotheses.

## Materials and methods

**Key resources table**

*Continued on next page*

*Continued*

| Reagent type (species) or resource | Designation | Source or reference | Identifiers | Additional information |
|---|---|---|---|---|
| Reagent type (species) or resource | Designation | Source or reference | Identifiers | Additional information |
| Genetic reagent (Pristionchus pacificus) | *Ppa-daf-6p::rfp* | this study | *tuEx231* | extrachromosomal array transgenic strain |
| Genetic reagent (Pristionchus pacificus) | *Ppa-daf-6p::venus* | this study | *tuEx250* | extrachromosomal array transgenic strain |
| Genetic reagent (Pristionchus pacificus) | *Ppa-odr-3p::rfp* | this study | *tuEx265* | extrachromosomal array transgenic strain |
| Genetic reagent (Pristionchus pacificus) | *Ppa-odr-7p::rfp* | this study | *tuEx296* and *tuEx297* | extrachromosomal array transgenic strain |
| Genetic reagent (Pristionchus pacificus) | *Ppa-che-1p::che-1:rfp* | this study | *lucEx367* | extrachromosomal array transgenic strain |
| Recombinant DNA reagent | *Ppa-che-1* mRNA probe | Stellaris, Biosearch Technologies; this study | PPA01143 | single-molecule in situ fluorescence probe |
| Sequence-based reagent | Ppa-daf-6 promoter forward primer | this study | PPA15978 | CTCGCCCGTGGATCATGTG |
| Sequence-based reagent | Ppa-daf-6 promoter reverse primer | this study | PPA15978 | TGCAAATCATTGAT TGAATCATGG |
| Sequence-based reagent | Ppa-odr-3 promoter forward primer | this study | PPA14189 | GAGCGAGTGAAATG AGCTCAGTCC |
| Sequence-based reagent | Ppa-odr-3 promoter reverse primer | this study | PPA14189 | GGGTGATCGATACGA GGAGTGTTC |
| Sequence-based reagent | Ppa-odr-7 promoter forward primer | this study | Contig1-aug1055.t1 | AACCAATGCATTGGCT TAGTTGGT TTCACTAATCACTACTG |
| Sequence-based reagent | Ppa-odr-7 promoter reverse primer | this study | Contig1-aug1055.t1 | CCCTTGTCATTCAGATGAGCGA GCTGATCAAGGAG |
| Sequence-based reagent | Ppa-che-1 promoter reverse primer | this study | PPA01143 | CAGGAAACAGCTATGACCATG |
| Sequence-based reagent | Ppa-che-1 intron reverse primer | this study | PPA01143 | CTGTGATAAGATCA TTATTGGTAC |
| Chemical compound, drug | DiI | Molecular Probe | V22889 | 1:150 dilution |
| Chemical compound, drug | DiO | Molecular Probe | V22886 | 1:150 dilution |
| Software, algorithm | TrakEM2 | PMID: 22723842 | | EM section alignment |
| Software, algorithm | 3D reconstruction | bioRxiv 485771 | | volumetric reconstruction |
| Software, algorithm | Photoshop | Adobe | | image processing |
| Software | Image J | Imagej.net | | image processing |
| Algorithm | www.phylogeny.fr | PMID: 18424797 | | maximum liklihood phylogeny |

## Transmission electron microscopy

The TEM data used in the current study are derived from two sets of slightly different preparations of young adult *P. pacificus* strain PS312 (*Sommer et al., 1996*) hermaphrodites. All of the 3D reconstructions and most descriptions of the morphological details originate from two datasets of roughly 3000 serial sections of 50 nm thickness, covering the anterior parts of two high-pressure-frozen and freeze-substituted adult hermaphrodite (specimen 107 and specimen 148), which were generated by Daniel Bumbarger. For a detailed methods description see Bumbarger and coworkers (*Bumbarger et al., 2013*). Alignment and manual segmentation was done in TrakEM2

(*Cardona et al., 2012*; *Kremer et al., 1996*). Segmentation and 3D reconstruction of the sensory head neurons were initially performed by Tahmineh Sarpolaki (2011) in the course of her Diploma thesis. The datasets can be accessed freely at https://wklink.org/7348.

Analysis of neuronal adjacencies was performed by using a modified python script written by Christopher Brittin (*Brittin et al., 2018*) Circuit diagrams were generated using Cytoscape (*Shannon et al., 2003*) and graphs were generated using the ggplot2 package for R (*Wickham, 2016*). In this study, as well as other comparisons of nematode electron micrographs, we find that smaller synapses show more variation. Thus, small synapses can be 'noisy' – this is likely due to biological noise but also some technical issues with evaluating small synapses.

We additionally used sections of four older TEM specimens, three transversely sectioned and one sagittally, which were prepared as follows: worms were placed in 100 µm deep specimen carriers half-filled with thick *E. coli* OP50 suspension, covered with the flat side of another carrier and high-pressure-frozen with a Bal-tec HPM-10 high-pressure freezer (Balzers, Liechtenstein). Freeze substitution was carried out in a freeze-substitution unit (Balzers FSU 010, Bal-Tec, Balzers, Liechtenstein) according to the following protocol: fix in 2% $OsO_4$, 0.5% UA, 0.5% GA in 97.5% acetone, 2.5% Methanol for 24 hr at −90°C, raise temperature to −60°C in 3 hr, hold for 6 hr, raise to −40°C in 2 hr, hold for 12 hr, keep on ice for 1 hr, wash with 100% acetone, embed in Epon/acetone. Blocks were sectioned with an LKB 2128 Ultratome. Ultrathin sections were viewed in a Philips CM10 or in a Fei Tecnai G2 Spirit T12 transmission electron microscope, images were acquired on photo plates or with a Morada TEM CCD camera, respectively.

## Scanning electron microscopy

Clean specimens of adult *P. pacificus* strain PS312 (*Sommer et al., 1996*) were fixed in 2.5% glutaraldehyde in PBS, post-fixed with 1% osmium tetroxide in PBS, dehydrated in a graded series of 30%, 50%, 70%, 95% and 100% ethanol, critical-point dried in liquid CO2 and sputter-coated with 10 nm Au/Pd. Inspection was carried out at 15 kV in a Hitachi S-800 field emission scanning electron microscope. We viewed ten processed *P. pacificus* PS312 hermaphrodite adults 'face on' and did not notice any differences in the external structures of the mouth regions among them.

## Transgenic reporter strains

*P. pacificus* California PS312 and transgenic strains were raised on OP50 *E. coli* seeded NGM plates at 20°C. We used *P. pacificus* gene names (PPAxxxxx) and putative homology from www.wormbase. org based on the most recent *P. pacificus* Genome Assembly El_Paco (*Rödelsperger et al., 2017*) and *C. elegans* WS269. To confirm the assigned gene orthology, we looked for best reciprocal BLASTP hits between *P. pacificus* and *C. elegans* genomes, as well as 1–1 orthology in gene phylogeny trees. To resolve ambiguities, we also performed BLASTP search against AUGUSTUS gene predictions in the previous *P. pacificus* Genome Assembly Hybrid1 (www.pristionchus.org >Genome > Hybrid1, select track AUGUSTUS2013) and the amino acid sequences downloaded under 'Sequences'. To make a *Ppa-odr-3* reporter plasmid, a ~ 1.7 kb long region upstream of the first ATG codon of *Ppa-odr-3* (PPA14189) (FP: GAGCGAGTGAAATGAGCTCAGTCC, RP: GGGTGATCGATACGAGGAGTGTTC) and the coding sequence of TurboRFP fused to the 3' UTR of the ribosomal gene *Ppa-rpl-23* (*Schlager et al., 2009*) were cloned into the pUC19 plasmid using Golden Gate Assembly Mix (New England BioLabs, E1600S) following the manufacturer's instructions. To make the *Ppa-daf-6* reporter, a 2.4 kb promoter sequence upstream of the start codon *Ppa-daf-6* (PPA15978) (FP: CTCGCCCGTGGATCATGTG, RP: TGCAAATCATTGATTGAATCATGG) was fused with *rfp* and *Venus* by fusion PCR (*Hobert, 2002*; *Kieninger et al., 2016*). Because the homology assignment for *odr-7* on Wormbase.org was *Ppa-nhr-66*, we looked for more likely orthology candidates using the AUGUSTUS genome assembly. The best BLASTP hit of *Cel-odr-7* was Contig1-aug1055.t1 (Contig1:2723982–2725788) and the *Ppa-odr-7* promoter was amplified and fused to *rfp* (FP: AACCAATGCATTGGCTTAGTTGGTTTCACTAATCACTACTG, RP: CCCTTGTCATTCAGATGAGCGAGCTGATCAAGGAG). To test if *Cel-oig-8* expression in the *Ppa-odr-7*-expressing neurons is sufficient to induce branching, we fused the 1.8 kb *Ppa-odr-7* promoter region to the genomic region of *Cel-oig-8* (FP: CCCTTGTCATTCAGATGAGCCTCCTTTCCAATATT, RP: TTACAGGGAGAAAGAGCATGTAG) and injected it with the *Ppa-odr-7p::rfp*. Each fusion junction site was verified by Sanger sequencing. For the construction of *Ppa-che-1p::rfp*, a 3.1 kb upstream fragment containing the first exon was

amplified and fused with *rfp*. Although *Ppa-che-1*(PPA01143) was not the best hit when using the *Cel-che-1* as query, *Cel-che-1* was the best hit when using PPA01143 as query. The best hit homolog on Wormbase.org, *Ppa-blmp-1*(PPA04978), is predicted to encode for a much larger protein than both *che-1* and *Ppa-che-1*(PPA01143). Curiously, no RFP fluorescence was visible when the reporter did not include the endogenous first exon and intron fused to *rfp*. The expression pattern observed with this transgene recapitulated the pattern of expression of the endogenous *Ppa-che-1* determined by single-molecule fluorescence in situ hybridization (smFISH). The probe set used to stain for *Ppa-che-1* was obtained from Stellaris, Biosearch Technologies (Middlesex, UK) and targets the validated, full-length *Ppa-che-1* sequence. The smFISH staining was performed as previously described (*Ji and van Oudenaarden, 2012*).

To create complex arrays for transgenesis, wildtype PS312 genomic DNA, the PCR product or the plasmid carrying the target reporter, and the *Ppa-egl-20::rfp* reporter (co-injection marker expressed in the tail) were digested with the FastDigest PstI restriction enzyme (Thermo Fisher Scientific, FD0615) and then mixed at the final concentration of 60 ng/μl for genomic DNA and 10 ng/μl for each plasmid. Prepared mix was injected in the gonad rachis in hermaphrodites (*Cinkornpumin and Hong, 2011*; *Schlager et al., 2009*). The $F_1$ progeny of injected animals were examined under a fluorescent dissecting microscope and animals that expressed the *Ppa-egl-20p::rfp* co-injection marker were isolated. All reporters were maintained as extrachromosomal arrays: *Ppa-daf-6p::rfp* (*tuEx231*), *Ppa-daf-6p::venus* (*tuEx250*), *Ppa-che-1p::rfp* (*lucEx367*), *Ppa-odr-3p::rfp* (*tuEx265*) and *Ppa-odr-7p::rfp* (*tuEx296* and *tuEx297*).

## DiI live staining

Lipophilic dyes DiI and DiO (Molecular Probes, V22889 and V22886) were used as neuronal tracers for a stereotypical subset of amphid neurons to facilitate cell identification either by relative cell position or overlap of red or green fluorescence in combination with transgenic reporters. DiI (red) and DiO (green) specifically stain five head amphid neurons in *C. elegans*, but DiO stains only two (AM2(ADL) and AM8(ASJ)) of the five (AM1(ASH), AM2(ADL), AM4(ASK), AM8(ASJ), and AM9(ADF)) possible stereotypical amphid neurons in *P. pacificus*. Well-fed nematodes were washed once in M9 buffer and then incubated for 2 hr at ~23°C with 300 μl of fresh M9 containing 1:150 dilution of DiI or DiO (6.7 μM). The nematodes were subsequently washed twice in 800 μl of fresh M9 buffer and placed onto OP50 *E. coli* seeded NGM plates to let the worms crawl freely for ~30 min to remove excess dye.

## Software

For images of DiI-filling in *Figure 3A–B*, we used AutoDeblur and Autovisualize v9.3 (AutoQuant Imaging, Inc, New York) to reduce fluorescence background, and MetaMorph v6.2r5 (Universal Imaging Corp. Pennsylvania) to 3D reconstruct or stack images from multiple planes. Adobe Photoshop and Image J were used to process other images.

## Phylogeny

The amino acid sequences of potential homologs were first identified by BLASTX searches on WormBase. The phylogeny trees were built using the following workflow: alignment and removal of positions with gap with T-COFFEE, Maximum Likelihood phylogeny by PhyML, and tree rendering by TreeDyn (www.phylogeny.fr) (*Dereeper et al., 2008*). Midpoint rooting was used and branch support ≥30% is shown. For the nuclear hormone receptor tree building, only the DNA binding C4 domain was used.

## Acknowledgements

We would like to thank Juergen Berger for assistance with scanning electron microscopy, Heinz Schwarz and Brigitte Sailer for assistance with transmission electron microscopy, Chris Crocker and Martin Voetsch for help with illustrations, Melissa Culhane, Suryesh Namdeo and Hanh Witte for technical assistance. We thank John White and Jonathan Hodgkin for the donation of the MRC/LMB electron microscopy archives to the Hall lab. RLH is supported by the National Institutes of Health Award SC3GM105579. SJC is supported by National Institutes of Health Fellowship

5F32MH1154328. OH is supported by the Howard Hughes Medical Institute. We also acknowledge the support of NVIDIA Corporation with the donation of a Titan V GPU used for this research.

## Additional information

### Competing interests

Oliver Hobert: Reviewing Editor, eLife. The other authors declare that no competing interests exist.

### Funding

| Funder | Grant reference number | Author |
| --- | --- | --- |
| National Institutes of Health | 5SC3GM105579 | Ray L Hong |
| National Institutes of Health | 5F32MH1154328 | Steven J Cook |

The funders had no role in study design, data collection and interpretation, or the decision to submit the work for publication.

### Author contributions

Ray L Hong, Conceptualization, Formal analysis, Supervision, Funding acquisition, Investigation, Visualization, Methodology, Writing—original draft, Project administration, Writing—review and editing; Metta Riebesell, Data curation, Formal analysis, Visualization, Methodology, Writing—original draft, Writing—review and editing; Daniel J Bumbarger, Heather R Carstensen, Luisa Cochella, Eduardo Moreno, Data curation, Formal analysis, Writing—review and editing; Steven J Cook, Data curation, Formal analysis, Investigation, Visualization, Writing—original draft, Writing—review and editing; Tahmineh Sarpolaki, Jessica Castrejon, Data curation, Formal analysis; Bogdan Sieriebriennikov, Conceptualization, Writing—original draft, Writing—review and editing; Oliver Hobert, Conceptualization, Resources, Supervision, Project administration, Writing—review and editing; Ralf J Sommer, Resources, Project administration, Writing—review and editing

### Author ORCIDs

Ray L Hong https://orcid.org/0000-0003-1870-8659
Steven J Cook https://orcid.org/0000-0002-1345-7566
Heather R Carstensen https://orcid.org/0000-0002-2679-3286
Luisa Cochella https://orcid.org/0000-0003-4018-7722
Eduardo Moreno http://orcid.org/0000-0001-7536-4122
Oliver Hobert https://orcid.org/0000-0002-7634-2854
Ralf J Sommer https://orcid.org/0000-0003-1503-7749

### Decision letter and Author response

Decision letter https://doi.org/10.7554/eLife.47155.052
Author response https://doi.org/10.7554/eLife.47155.053

## Additional files

### Supplementary files

• Transparent reporting form
DOI: https://doi.org/10.7554/eLife.47155.050

### Data availability

All data generated or analysed during this study are included in the manuscript and supporting files. Source files have been provided for Figure 9. Our electron micrograph dataset is available on web-Knossos (https://wklink.org/7348).

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
