## [Decision Letter]

Thank you for submitting your article "Evolution of neuronal anatomy and circuitry in two highly divergent nematode species" for consideration by *eLife*. Your article has been reviewed by three peer reviewers, and the evaluation has been overseen by a Reviewing Editor and Eve Marder as the Senior Editor. The reviewers have opted to remain anonymous.

The reviewers have discussed the reviews with one another and the Reviewing Editor has drafted this decision to help you prepare a revised submission.

Summary:

All reviewers found the work of interest but had several issues that require clarification before the manuscript can be accepted.

Essential revisions:

1) Subsection “Patterns of neuronal dye-filling are largely conserved”: The description of dye-filling is confusing. The neurons stained by DiO in *C. elegans* are neither mentioned or shown. A list of the neurons stained by DiO in *C. elegans*, and if possible an image of DiO staining, would be very helpful. In addition, please list the DiI-stained neurons in *P. pacificus* in the same order as you list the DiI-stained neurons in *C. elegans* so that it is more readily apparent to the reader which neurons are different between the two species. It would also be helpful to explicitly state the differences in which neurons stain between the two species (i.e., which neurons stain in both species, which neurons stain in only one), rather than saying that staining is "similar" in *P. pacificus* (since the sets are overlapping but not exactly the same). Also indicate differences between DiO and DiI staining.

2) Please add a table showing the synaptic connectivity of *P. pacificus* (similar to what is provided on wormwiring.org for Ce). It's difficult to extract the full connectivity data from the graph shown in Figure 8B (although the graph is useful). The connectivity data do not appear to be represented anywhere else in the manuscript, and so a list of connections would be extremely helpful. Without this, it's not clear how other readers will be able to access this data. Consider also showing the connectivity network obtained even without filtering out the weaker synapses (or reduce the stringency of your filtering). It may reveal more species-specific synapses.

3) The conclusions of the paper are very similar to a previous paper by some of the same authors describing the pharyngeal nervous system in *P. pacificus*. This paper also concluded that the cell body positions are conserved and it is possible to establish cell homologies, however, neuronal connectivity is much less conserved. The present paper also reports changes in sensory morphologies. The authors need to discuss clearly how this work builds on and relates to the 2013 paper. Please also refer to additional relevant work: The conclusion that the ancestral state of ASE may have been symmetric was brought up in Etchberger et al., 2009 ("The two ASE neurons may have initially been identical, with CHE-1 controlling the exact same set of genes in ASEL and ASER.") and relates to similar conclusions about symmetric vs. lateralized expression of gcy-4 in C. briggsae vs. *C. elegans* from Ortiz et al., 2006. As a more minor example, DiI staining patterns of *P. pacificus* (along with proposed homology assignments) were reported in Srinivasan et al., 2008 but those interpretations are not directly cited or compared with the current results.

4) It is not clear how the authors intend to share the raw EM image stacks and the corresponding neuron reconstructions. Without the full release of all data and traces in a user-friendly format (e.g. the TrakEM databases), it would be very difficult to query these data by the specialists or anyone interested in neuron morphology and connectomes. Please provide a plan for data sharing.

5) The authors need to emphasize that the assignation of homology was made using different sources of evidence for different cells. Ideally, criteria independent of the features whose evolution is being analyzed (cilia morphology, gene expression) should be used for determining homology. Otherwise, circular arguments may arise. It may be thus safer to first assign homology only on cell, position, axonal projection, and dye uptake (and justify the use of these criteria). Then, cilia morphology, gene expression and connectivity can be mapped onto the assignments and analyze what is similar and what is different. Additional column(s) in Table 2 could indicate alternative assignments, level of confidence in the assignment, and/or main differences between the proposed homologs (e.g. uniciliated vs. biciliated ADL). The column “Notable features" in Table 2 could be renamed to “Features supporting homology assignment". Also modify the text to more rigorously define the criteria for homology assignments and to explain what evidence would have been needed to reject the hypothesis that some neuron types have been gained or lost instead of always having a one-to-one homology. For example, no reasoning is given for the assignment of AS10 as ASI, or AS11 as AWB. As another example, AD3 is assigned as AWA yet there is no positive support for this assignment and three lines of evidence that tend to refute it (cilia morphology, odr-3 expression, odr-7 expression). Finally, if there is any information regarding neurotransmitter identities – please provide these to strengthen the homology assignments.

6) It would be useful to speculate in brief in the Discussion regarding the functional consequences of the observed differences especially with regards to the different lifestyle of Pp compared to Ce. In particular, it would be useful to include some discussion on whether there is functional relevance to the divergent morphologies or whether they only came about by evolutionary drift, maybe through 'network systems drift' similar to the developmental systems drift that has been reported for some of the developmental pathways of Pristionchus.

7) If possible, please provide a brief discussion of whether the newly described morphologies and connections represent the derived or the ancestral condition.

8) The findings need to also be discussed in the broader context of the evolution of neural circuits and sensory systems. For example, seminal work on *Drosophila* chemosensation, where sensory receptor evolution could be linked with species-specific ecologies and changes in neuroanatomy are not discussed (e.g. work from the Hansson and Benton labs). The pioneering work of Paul Katz on locomotor circuits in nudibranchs could also be discussed. This work demonstrated the conservation of neuron positions and identities and the rapid evolution of connectivity between homologous neurons. In nudibranchs, this could be integrated with function through detailed electrophysiological recordings across species.

9) Include the methods used to reconstruct the phylogenetic trees. The trees lack any indication of statistical support.

10) Schematics showing the overview of the reconstructed cilia would be very useful and would show the striking difference in cilia morphology. One to one comparisons to *C. elegans* if that is possible.

11) Are the *P. pacificus* specific synaptic connections present in both specimens analyzed? Do you see more variation in less conserved synapses?

12) "Proportion of adjacency" is defined as the "total number of EM sections where two processes are adjacent divided by the number of sections in the region of interest" yet it is sometimes >100%. It would help to explain this and to provide more detail on the method (the Brittin et al., 2018 pre-print that is cited does not seem to use proportion of adjacency). Also, the discussion of neuronal neighborhoods and synaptic connectivity would benefit from citing White et al., 1983 and comparing the current results to this analysis. Is "proportion of adjacency" here the same as the previously used "relative adjacency"? Their major conclusion was that, while clearly some specificity mechanisms are in place, neurons make synapses with many of their potential partners (520 of 1165 neighbor pairs, or 45%, had synapses). How does this compare to the current measurements?

[Editors' note: further revisions were requested prior to acceptance, as described below.]

Thank you for submitting your article "Evolution of neuronal anatomy and circuitry in two highly divergent nematode species" for consideration by *eLife*. Your article has been reviewed by three peer reviewers, and the evaluation has been overseen by a Reviewing Editor and Eve Marder as the Senior Editor. The following individual involved in review of your submission has agreed to reveal their identity: Gáspár Jékely (Reviewer #2).

The reviewers have discussed the reviews with one another and the Reviewing Editor has drafted this decision to help you prepare a revised submission.

Summary:

All reviewers thought that the revised manuscript was much improved, that the paper was much stronger, and that this would be a valuable resource.

Essential revisions:

However, all reviewers still had issues regarding the circular reasoning used for assigning homologies. For instance, features like soma position, synaptic connectivity and dye-filling are used to assign homology but then it is argued that these features are conserved. Thus, assumptions are conflated with conclusions.

1) It is recommended that the manuscript include a section in the Results that explains the methodology used to make homology assignments. This should also explicitly address the circular reasoning used, cite precedent(s) for this if applicable, address the limitations of this approach in the absence of comparative functional data, and clearly state the level of confidence for homology assignment for each neuron type.

2) Also explain the evidence that would be needed to conclude that 1-to-1 homology assignments are justified as compared to potential gain or loss of specific neuron types.

---

## [Author Response]

Essential revisions:1) Subsection “Patterns of neuronal dye-filling are largely conserved”: The description of dye-filling is confusing. The neurons stained by DiO in *C. elegans* are neither mentioned or shown. A list of the neurons stained by DiO in C. elegans, and if possible an image of DiO staining, would be very helpful. In addition, please list the DiI-stained neurons in P. pacificus in the same order as you list the DiI-stained neurons in C. elegans so that it is more readily apparent to the reader which neurons are different between the two species. It would also be helpful to explicitly state the differences in which neurons stain between the two species (i.e., which neurons stain in both species, which neurons stain in only one), rather than saying that staining is "similar" in P. pacificus (since the sets are overlapping but not exactly the same). Also indicate differences between DiO and DiI staining.

We concur that a side-by-side comparison of live dye filled amphid neurons between *C. elegans* and *P. pacificus* would be informative, so we have now added panels in Figure 4 showing five DiO stained amphid neurons in *C. elegans*, in comparison to only two DiO stained amphid neurons in *P. pacificus* (Panels I and I’). We also significantly revised Table 2 to compare dye filling features for each nominated homolog between *C. elegans* and *P. pacificus*. In addition, a side-by-side comparison of dye-filling property between the two species is now summarized in the new Table 3. We now also introduce in the dye-filling section the *P. pacificus* DiI staining pattern in the same order as we listed the DiI-stained neurons in *C. elegans*.

2) Please add a table showing the synaptic connectivity of P. pacificus (similar to what is provided on wormwiring.org for Ce). It's difficult to extract the full connectivity data from the graph shown in Figure 8B (although the graph is useful). The connectivity data do not appear to be represented anywhere else in the manuscript, and so a list of connections would be extremely helpful. Without this, it's not clear how other readers will be able to access this data. Consider also showing the connectivity network obtained even without filtering out the weaker synapses (or reduce the stringency of your filtering). It may reveal more species-specific synapses.

We completely agree that providing such data is important. To this end, we have included three.csv files that include the connectivity information by neuronal class (Interneurons and other amphid neurons in Figure 9—source data 1-3). Figure 9—source data 3 lists synapses present in one species but not the other that are less than 10 sections in synaptic strength. The stringency we used for determining a species-specific synapse was based upon the cutoff for a sex-specific connection from Cook et al., 2019 (Nature, in press). We decided to use this conservative cutoff given the small number of samples reconstructed.

3) The conclusions of the paper are very similar to a previous paper by some of the same authors describing the pharyngeal nervous system in P. pacificus. This paper also concluded that the cell body positions are conserved and it is possible to establish cell homologies, however, neuronal connectivity is much less conserved. The present paper also reports changes in sensory morphologies. The authors need to discuss clearly how this work builds on and relates to the 2013 paper.

We were very torn in our original submission whether and how to include this past work. The reason for this is that currently unpublished work (https://search.proquest.com/docview/1969077248?pq-origsite=gscholar) strongly suggests that the cellular basis for synaptic specificity in the pharyngeal nervous system is fundamentally distinct from that of the extra-pharyngeal nervous system, which leads to a very different interpretation of the synaptic differences in *P. pacificus* and *C. elegans* observed in the pharynx in the past study and in the amphid sensory system, as presented in this manuscript. Based on the reviewers’ comments, we understand that this was too glaring an omission and we now do discuss these differences in the Conclusions of the paper. The most relevant point is that the findings presented in this paper significantly extend the conclusions of the previous Cell paper.

Please also refer to additional relevant work: The conclusion that the ancestral state of ASE may have been symmetric was brought up in Etchberger et al., 2009 ("The two ASE neurons may have initially been identical, with CHE-1 controlling the exact same set of genes in ASEL and ASER.") and relates to similar conclusions about symmetric vs. lateralized expression of gcy-4 in *C. briggsae* vs. *C. elegans* from Ortiz et al., 2006.

We have added in the Conclusions the work by Etchberger et al., 2009 that suggests ASE laterality likely arose from a bilaterally symmetric ground state.

As a more minor example, DiI staining patterns of P. pacificus (along with proposed homology assignments) were reported in Srinivasan et al., 2008 but those interpretations are not directly cited or compared with the current results.

The DiI staining pattern found in this study is seemingly identical to that reported by Srinivasan et al., 2008, but the Srinivasan study made the homology assignments based on the assumption that cell positions visible by DIC are conserved between *C. elegans* and other nematodes species, including *P. pacificus*. However, based on new criteria available through EM data, we can only confirm the ASJ and ASH neurons to be dye filled and in the same positions in both species, but argue that the ASK, ADL, ASI cells in *P. pacificus* occupy different cell positions. We have now interpreted our findings in the context of the previous Srinivasan study.

4) It is not clear how the authors intend to share the raw EM image stacks and the corresponding neuron reconstructions. Without the full release of all data and traces in a user-friendly format (e.g. the TrakEM databases), it would be very difficult to query these data by the specialists or anyone interested in neuron morphology and connectomes. Please provide a plan for data sharing.

We are working with www.webKnossos.org to provide direct link to our dataset, where researchers would be able to freely view and analyze the data. webKnossos is a browser-based annotation tool for 3D EM datasets developed in collaboration with the Max-Planck Institute for Brain Research. One of the co-authors currently has his EM dataset hosted at webKnossos (Bumbarger et al., 2013). This dataset reference has been added to the EM section of Materials and methods.

5) The authors need to emphasize that the assignation of homology was made using different sources of evidence for different cells. […] Finally, if there is any information regarding neurotransmitter identities – please provide these to strengthen the homology assignments.

These are very useful suggestions. Hence, we have significantly revised Table 2 to better categorize independent features that support homology (cell position, dye filling, axon termination site, amphid interneuron synaptic targets) versus features that may have evolved (cilia morphology, gene expression). In this revised table, AWA homology is indeed the weakest and assigned due to the other neurons having positive support. Although serotonin immune-reactive neurons have been detected in the head neurons in *P. pacificus* (Loer and Rivard, 2007), it is not clear which amphid homologs they might be. We hope our work will be able to promote future studies into neurotransmitter identities so that they could be mapped onto the amphid neuron identities featured in this paper.

6) It would be useful to speculate in brief in the Discussion regarding the functional consequences of the observed differences especially with regards to the different lifestyle of Pp compared to Ce. In particular, it would be useful to include some discussion on whether there is functional relevance to the divergent morphologies or whether they only came about by evolutionary drift, maybe through 'network systems drift' similar to the developmental systems drift that has been reported for some of the developmental pathways of Pristionchus.

We have now added a final paragraph in the Conclusions regarding changes in developmental systems. Since we are in the early stages of outlining divergent gene expressions and cell morphologies, follow-up studies could allow us to investigate if these changes are due to “network systems drift.”

7) If possible, please provide a brief discussion of whether the newly described morphologies and connections represent the derived or the ancestral condition.

We have added Figure 1—figure supplement 3, which maps select sensory neurons onto *P. pacificus, C. elegans*, and *A. complexus* phylogeny. This comparison shows that wing neurons and unciliated URX seem to be derived characters in *C. elegans*, while ciliated BAG, unciliated URY, and short processes in AUA are conserved.

8) The findings need to also be discussed in the broader context of the evolution of neural circuits and sensory systems. For example, seminal work on *Drosophila* chemosensation, where sensory receptor evolution could be linked with species-specific ecologies and changes in neuroanatomy are not discussed (e.g. work from the Hansson and Benton labs). The pioneering work of Paul Katz on locomotor circuits in nudibranchs could also be discussed. This work demonstrated the conservation of neuron positions and identities and the rapid evolution of connectivity between homologous neurons. In nudibranchs, this could be integrated with function through detailed electrophysiological recordings across species.

We thank the reviewer for alerting us to the Katz papers which we now include, it indeed raises interesting perspectives on our own work. As requested, we also now discuss work in insects (incl. Benton labs) – which actually very nicely allows us to emphasize the novelty of our work: In insects, evolutionary novelties in the olfactory system arise (among other mechanisms) through differences in olfactory neuron targeting properties (i.e. projections to distinct glomeruli). In contrast, we reveal here striking constraints in axon patterning and projections, i.e. neighborhoods of axons are completely constrained – but the novelty lies in choosing distinct synaptic partners within these constrained neighborhoods. We highlight this now in the Discussion.

9) Include the methods used to reconstruct the phylogenetic trees. The trees lack any indication of statistical support.

The cladograms have now been replaced or enhanced with Maximum Likelihood phylogeny trees with branch support. Due to computational constraint with the tree-building platform, we only reanalyzed the taste ASE GCYs in Figure 4—figure supplement 4, which is mostly congruent with the original cladogram in Supplementary Figure 1A. The amino acid sequences of potential homologs were first identified by BLASTX searches on WormBase. The phylogeny trees were built using the following workflow: alignment and removal of positions with gap using T-COFFEE, Maximum Likelihood phylogeny by PhyML, and tree rendering by TreeDyn (www.phylogeny.fr). The method for tree construction has been added to the Materials and methods section and updated in the corresponding figure legend.

10) Schematics showing the overview of the reconstructed cilia would be very useful and would show the striking difference in cilia morphology. One to one comparisons to *C. elegans* if that is possible.

We added a new Figure 5 to make one-to-one comparisons of the morphologies of the homologous neurons between the two species.

11) Are the P. pacificus specific synaptic connections present in both specimens analyzed? Do you see more variation in less conserved synapses?

In this study, as well as other comparisons of nematode electron micrographs, we find that smaller synapses show more variation. We believe that small synapses can be ‘noisy’ – this is likely due to biological noise but also some technical issues with evaluating small synapses. This justification is now noted in the text. Nevertheless, we did evaluate connectivity in both series 14 and series 15 and found that larger *Pristionchus pacificus*-specific synapses are present in both series. A future publication will investigate the variability between samples in greater detail.

12) "Proportion of adjacency" is defined as the "total number of EM sections where two processes are adjacent divided by the number of sections in the region of interest" yet it is sometimes >100%. It would help to explain this and to provide more detail on the method (the Brittin et al., 2018 pre-print that is cited does not seem to use proportion of adjacency). Also, the discussion of neuronal neighborhoods and synaptic connectivity would benefit from citing White et al., 1983 and comparing the current results to this analysis. Is "proportion of adjacency" here the same as the previously used "relative adjacency"? Their major conclusion was that, while clearly some specificity mechanisms are in place, neurons make synapses with many of their potential partners (520 of 1165 neighbor pairs, or 45%, had synapses). How does this compare to the current measurements?

For proportion of the nerve ring, the numerator is the number of EM sections where two processes are adjacent, and the denominator is the total number of EM sections within the region of interest. 100% adjacent would suggest that these two processes were adjacent for the entire ipsilateral region of interest. As some neurons project to the contralateral side of the nerve ring, it is possible for them to be adjacent for greater than 100% of nerve ring. We chose this designation because most amphid neurons terminate at the dorsal midline.

Our definition is similar to that of White et al., 1983, which is now cited in the text. White et al., 1983 wrote ‘The degree of adjacency of each neighbor was determined by counting the number of sections in which it was adjacent to the process of the neuron under consideration. The numbers thus obtained were normalized to the neighbor with the greatest adjacency. The data thus obtained were plotted in tabular form along with the synaptic data. We have selected a few examples of neuron classes that illustrate most of the features that were seen.’ In our analysis we did not scale to the strongest neighbor. We also chose not to use the word ‘degree’ in the definition due its definition and use in the graph theory literature.

The findings in this paper, and in a more detailed comparison, Brittin et al., 2019 (in revision), suggest that a similar proportion of neurons make a synapse with the neighbors. However, our analyses were done slightly differently from White and colleagues. Their analysis of adjacency was limited to one sample (JSH series, an L4 hermaphrodite), and included analyzing synaptic partners manually every 5^th^ EM section. Our method was computational, for every section, leading to increased sensitivity and accuracy. We are therefore slightly hesitant to make any hard judgment compared to the original analysis.

[Editors' note: further revisions were requested prior to acceptance, as described below.]

Essential revisions:However, all reviewers still had issues regarding the circular reasoning used for assigning homologies. For instance, features like soma position, synaptic connectivity and dye-filling are used to assign homology but then it is argued that these features are conserved. Thus, assumptions are conflated with conclusions.1) It is recommended that the manuscript include a section in the Results that explains the methodology used to make homology assignments. This should also explicitly address the circular reasoning used, cite precedent(s) for this if applicable, address the limitations of this approach in the absence of comparative functional data, and clearly state the level of confidence for homology assignment for each neuron type.

We absolutely agree that the criteria for making the homology assignments should be stated more explicitly in the beginning of the Results and have done so. Homology assignments for the amphid neurons of other nematode species have relied primarily on conservation of cell body positions and morphology, so we have been more demanding in our own criteria by using additional characters (axon projections, dye filling, manner of dendrite entry into sheath, connection to first layer interneurons). We also added a new section at the end of the Results, “Uncertainties of homology assignments,” to describe the limitations of our approach as well as additional functional studies and cell lineage analysis that would resolve those uncertainties.

2) Also explain the evidence that would be needed to conclude that 1-to-1 homology assignments are justified as compared to potential gain or loss of specific neuron types.

Cellular homologies between individual blastomeres in the embryo and vulval precursor cells in the larva of *P. pacificus* and *C. elegans* suggest that our proposed 1-to-1 homologies of the mature amphid neurons could be validated by tracing the neuronal lineages of *P. pacificus* and examining the concordance of cell position (as we defined here) with cellular lineage. We suggested this approach as a future direction in the new Results section: “Uncertainties of homology assignments.”